# Nuclear myosin 1 activates *p21* gene transcription in response to DNA damage through a chromatin-based mechanism

Tomas Venit[1], Khairunnisa Semesta [1], Sannia Farrukh[1], Martin Endara-Coll [1,3], Robert Havalda[2,4], Pavel Hozak[2,4] & Piergiorgio Percipalle[1,3 ✉]

Nuclear myosin 1 (NM1) has been implicated in key nuclear functions. Together with actin, it has been shown to initiate and regulate transcription, it is part of the chromatin remodeling complex B-WICH, and is responsible for rearrangements of chromosomal territories in response to external stimuli. Here we show that deletion of NM1 in mouse embryonic fibroblasts leads to chromatin and transcription dysregulation affecting the expression of DNA damage and cell cycle genes. NM1 KO cells exhibit increased DNA damage and changes in cell cycle progression, proliferation, and apoptosis, compatible with a phenotype resulting from impaired p53 signaling. We show that upon DNA damage, NM1 forms a complex with p53 and activates the expression of checkpoint regulator p21 (*Cdkn1A*) by PCAF and Set1 recruitment to its promoter for histone H3 acetylation and methylation. We propose a role for NM1 in the transcriptional response to DNA damage response and maintenance of genome stability.

[1] Science Division, Biology Program, New York University Abu Dhabi (NYUAD), P.O. Box 129188 Abu Dhabi, UAE. [2] Department of Biology of the Cell Nucleus, Institute of Molecular Genetics, AS CR, v.v.i., Videnska 1083, 142 20 Prague, Czech Republic. [3] Department of Molecular Biosciences, The Wenner-Gren Institute, Stockholm University, SE-106 91 Stockholm, Sweden. [4] Laboratory of Epigenetics of the Cell Nucleus, Institute of Molecular Genetics of the ASCR, Division BIOCEV, Prague, Czech Republic. ✉email: pp69@nyu.edu

C ytoskeletal proteins such as actin and myosin have emerged as key factors in the regulation of a dynamic chromatin landscape compatible with active transcription[1]. Actin and a specific isoform of myosin 1c referred to as nuclear myosin 1 (NM1) associate with the genome in both insects and mammals[2–4]. Actin and NM1 synergize with all three nuclear RNA polymerases and chromatin remodelers to initiate and activate transcription[5,6]. NM1 interacts with the polymerase-associated actin and directly binds to the chromatin through its C-terminal domain[2,5,7]. NM1 is also a bona fide component of the large multiprotein assembly B-WICH, containing the chromatin remodeling complex WICH, with the subunits WSTF and the ATPase SNF2h[8]. NM1 binding to either polymerase-associated actin or the WICH complex depends on its motor function and for this reason NM1 has been suggested to function as a molecular switch[9]. Association of NM1 with the WICH complex is required for nucleosome repositioning activity and facilitates recruitment of the histone acetyltransferase (HAT) PCAF and histone methyl transferase (HMT) Set1B[2,8]. This recruitment directly leads to histone H3 acetylation and methylation and, therefore, allows for transcriptional activation[2,5]. However, it is currently not known whether the role of NM1 in establishing correct epigenetic signatures during transcription is also required in the transcriptional response to DNA damage.

The DNA damage response (DDR) consists of signaling cascades and DNA repair processes, which ultimately recognize and remove DNA lesions[10]. Chromatin regulation plays an essential role, as it has to be reorganized to allow DNA repair proteins to sites of DNA resection and fix the damage[11]. This entails chromosomal movements and repositioning of DNA breaks from highly compact nuclear territories to less condensed regions and vice versa. Recently, several myosin species, in conjunction with actin, have been shown to play a role in chromosomal movements in response to DNA damage. In *Drosophila*, depletion of Myo1A, Myo1B, and Myosin V resulted in a defective relocalization of heterochromatic double-strand breaks to the nuclear periphery[12]. Similarly, NM1 has been shown to play a role in rapid and actin-dependent chromosome territory relocation[13] and formation of contacts between homologous chromosomes after DNA damage[14]. Tight chromatin regulation is also necessary in the DNA-damage signaling cascade where chromatin remodeling and epigenetic mechanisms ensure transcription of DNA repair proteins and many signaling proteins important for cell cycle regulation. In the presence of double-strand breaks, the transcription factor p53 undergoes posttranslational modifications[15] and induces cell cycle arrest at the G1/S checkpoint by promoting activation of target genes such as *p21* (*Cdkn1A*) gene encoding the p21 (WAF1, Cip1) protein. Once activated, p21 binds to cyclin E and Cyclin A/CDK2 required for the G1/S-phase transition and inhibits their activity, thereby contributing to G1-phase arrest[16]. p21 can also directly regulate the expression of DNA damage-related genes by inhibition of histone-acetyltransferase p300 and downregulation of histone H4 acetylation at DDR gene promoters[17]. Indeed, p21-null human fibroblasts are more sensitive to DNA damage; they are deficient in DNA repair and are unable to arrest in G1 in response to DNA breaks[18]. Therefore, lack of p21 may induce tumorigenesis[19]. However, how *p21* gene activity is regulated at the chromatin level is not entirely understood.

In the present study, we set out to investigate the potential role of NM1 in the transcriptional response to DNA damage. We found that NM1 is directly involved in the regulation of *p21* gene activation. Using embryonic fibroblasts from an NM1-knockout (KO) mouse, we demonstrate that loss of NM1 leads to constitutive DNA damage. In line with these observations, NM1 KO mouse embryonic fibroblasts (MEFs) show higher proliferation

rates, increased γ-H2AX foci, and gene expression profiles obtained by RNA sequencing (RNA-Seq) corresponding to a p21 mutant phenotype. In addition, chromatin immunoprecipitation sequencing (ChIP-seq) and ChIP quantitative PCR (qPCR) experiments show that NM1 is enriched at the transcription start site (TSS) of the *p21* gene and occupancy is enhanced upon DNA damage. In MEFs subjected to NM1 knockdown (KD) by small interfering RNA (siRNA), p21 expression is significantly downregulated and we show that this is directly caused by impaired recruitment of the HAT PCAF and the HMT Set1 with loss of H3 acetylation and methylation. We propose a new role for NM1 in the transcriptional response to DNA damage through a chromatin-based mechanism.

## Results

**Epigenetic signatures and global transcription are altered in the absence of NM1.** Previous studies have shown that NM1 distribution across the mammalian genome correlates with RNA Polymerase II and active epigenetic marks at TSS of class II promoters[2]. To test whether NM1 affects the distribution of histone marks, we performed high-content phenotypic profiling of primary MEFs derived from NM1 wild-type (WT) and KO embryos (Supplementary Fig. 1a). Cells were stained with antibodies against epigenetic marks for constitutive heterochromatin (H3K9me3), active enhancers (H3K27ac and H3K4me1), and euchromatin (H3K9ac and H3K4me3) (Fig. 1a). Staining was quantified by using the Compartmental Analysis BioApplication software inbuilt in the High Content Screening platform and at least 10,000 cells were used for each measurement (Fig. 1b). Except for the repressive mark H3K9me3 whose levels increased in NM1 KO cells, we found significant drops in the levels of each of the active epigenetic marks tested in KO cells compared with WT (Fig. 1a, b). Results from western blotting analysis with the same antibodies correlate with the data obtained from high-content phenotypic profiling (Fig. 1c, d and Supplementary Fig. 4).

Changes in epigenetic signatures for both repressive and active epigenetic marks indicate that loss of NM1 may induce both gene activation and repression. To test this hypothesis, we studied the effect of NM1 depletion on transcription by RNA-Seq. For this analysis, we isolated total RNA from NM1 WT and KO primary MEFs. Three biological replicates in each experimental group were used for isolation of RNA, library preparation, and subsequent analysis. Principal component analysis plot and distance heatmap show that the two sets of samples cluster apart, indicating higher variability between experimental groups than between samples within the same groups (Supplementary Fig. 1b, c). To visualize the data, the MA plot was obtained using the differences between the gene expression in KO and WT MEFs (Supplementary Fig. 1d). The results from the deep sequencing were visualized by plotting the log2 (fold change between KO and WT condition) vs. the mean of normalized counts. Most of the genes do not show any significant change in their expression (marked grey). Among the differentially expressed genes (marked red), we found similar numbers of significantly upregulated (1258) and downregulated genes (817). This is compatible with significant alterations of both repressive and active epigenetic marks observed in NM1 KO cells (Supplementary Fig. 1d). Next, we performed Gene Ontology (GO) analysis on genes differentially expressed between KO and WT conditions. For this, we selected all genes with at least twofold difference between WT and KO cells with adjusted *p*-value < 0.05 and analyzed them using DAVID Bioinformatics software (https://david.ncifcrf.gov/). The most abundant GO terms found in the biological process category correlate with the previously described function of NM1 in Pol I

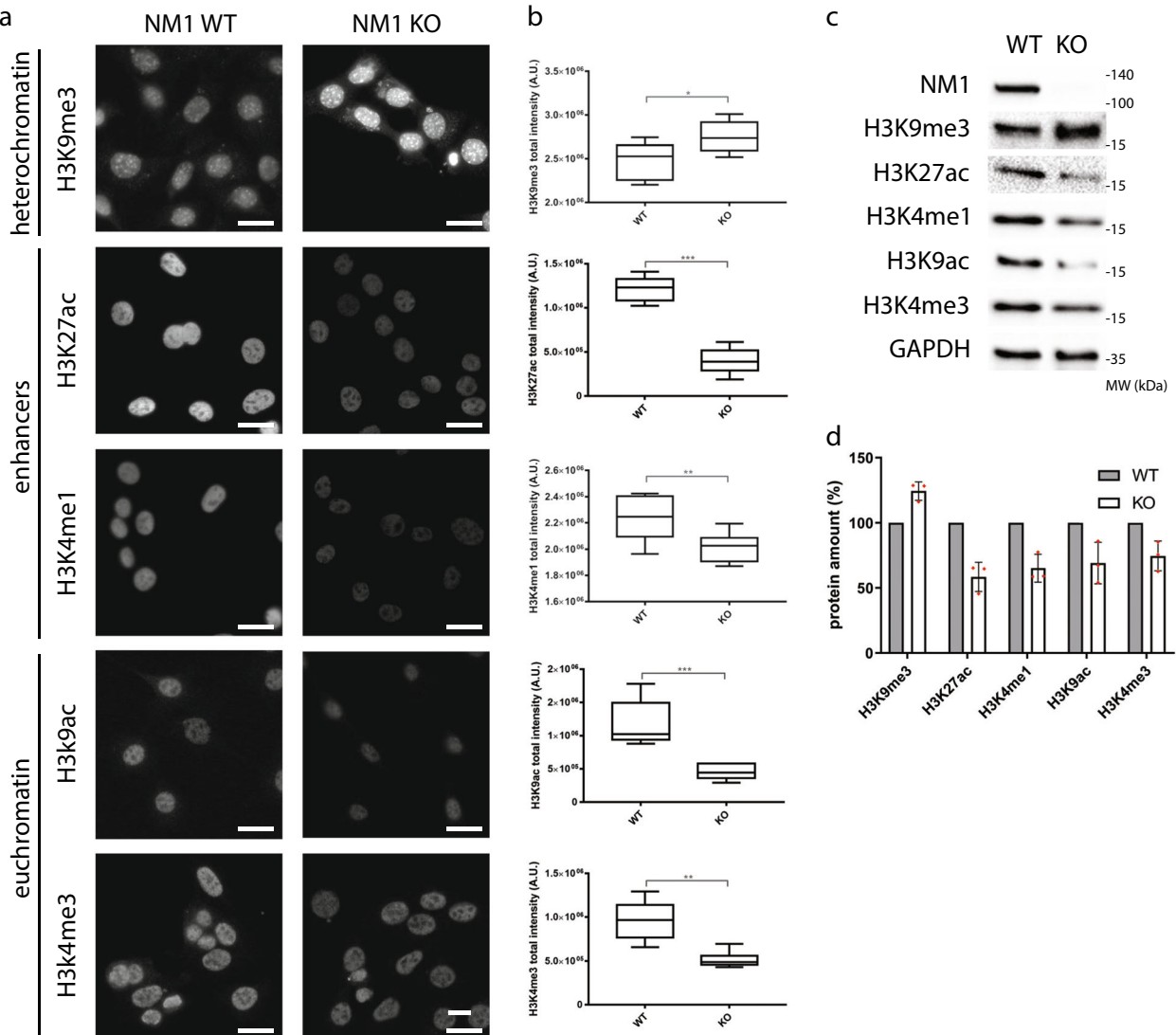

**Fig. 1 Histone epigenetic signatures are altered in the absence of NM1. a** NM1 WT and KO cells were immunostained with antibodies against different histone marks specific for heterochromatin (H3K9me3), euchromatin (H3K9ac and H3K4me3), and gene enhancers (H3K27ac and H3K4me1). Representative pictures for each staining are showed. Scale bar is 5 µm. **b** Nuclear staining intensity was quantified by high-content phenotypic profiling. Each box plot represents mean value and first and third quartile values. Error bars represents minimum and maximum values. For each measurement, at least 10,000 nuclei have been measured. $*p < 0.05$, $**p < 0.01$, $***p < 0.001$; ns (not significant), $n = 6$. **c** Representative western blottingss stained with antibodies against different histone marks, NM1, and control GAPDH. **d** Quantification of immunoblot staining in NM1 KO cells normalized to expression of each histone mark in WT cells and to GAPDH expression, $n = 3$.

and Pol II transcription (Fig. 2 and Supplementary Table 1). Compatible with this observation, NM1 ChIP-seq reads plotted within ±5 kb around TSS of genes with at least a twofold difference between WT and KO show that NM1 is enriched at TSS (Supplementary Fig. 3a–3d). Importantly, we found two groups of genes overrepresented, those related to cell cycle progression and those related to DNA-damage signaling and repair.

Therefore, we conclude that in the absence of NM1, cells are transcriptionally reprogramed, and that specialized cellular functions such as DNA-damage signaling and DNA repair are likely to be dysregulated.

**NM1 deletion leads to increased DNA damage, apoptosis, and cell proliferation rates.** As many genes related to DNA damage repair and signaling are dysregulated in the NM1 KO cells, we reasoned that there might be a higher occurrence of DNA damage

in NM1 KO MEFs. To address this question, we immunostained WT and KO primary MEFs with an anti-γH2AX antibody recognizing sites of DNA damage (Fig. 3a) and quantified it as mentioned before. Results from these experiments show that the number, intensity, and the total area of γH2AX-positive DNA damage foci are significantly higher in NM1 KO MEFs ($p < 0.001$) in comparison with WT MEFs (Fig. 3b). Compatible with increased γH2AX-positive foci, results from comet assays performed on both WT and NM1 KO MEFs revealed that, in the absence of NM1, there is a significant increase in the extent of DNA damage compared with the WT condition (Fig. 3c, d).

To exclude the possibility that the observed differences between NM1 WT and KO primary cells are caused by some spontaneous mutations of primary cells rather than NM1 deletion itself, we used Crispr/Cas9-editing system to generate stable NM1 KO cells (NM1 KO cMEFs). Results from immunostaining (Supplementary Fig. 2a, b) and western blot analysis with anti-γH2AX

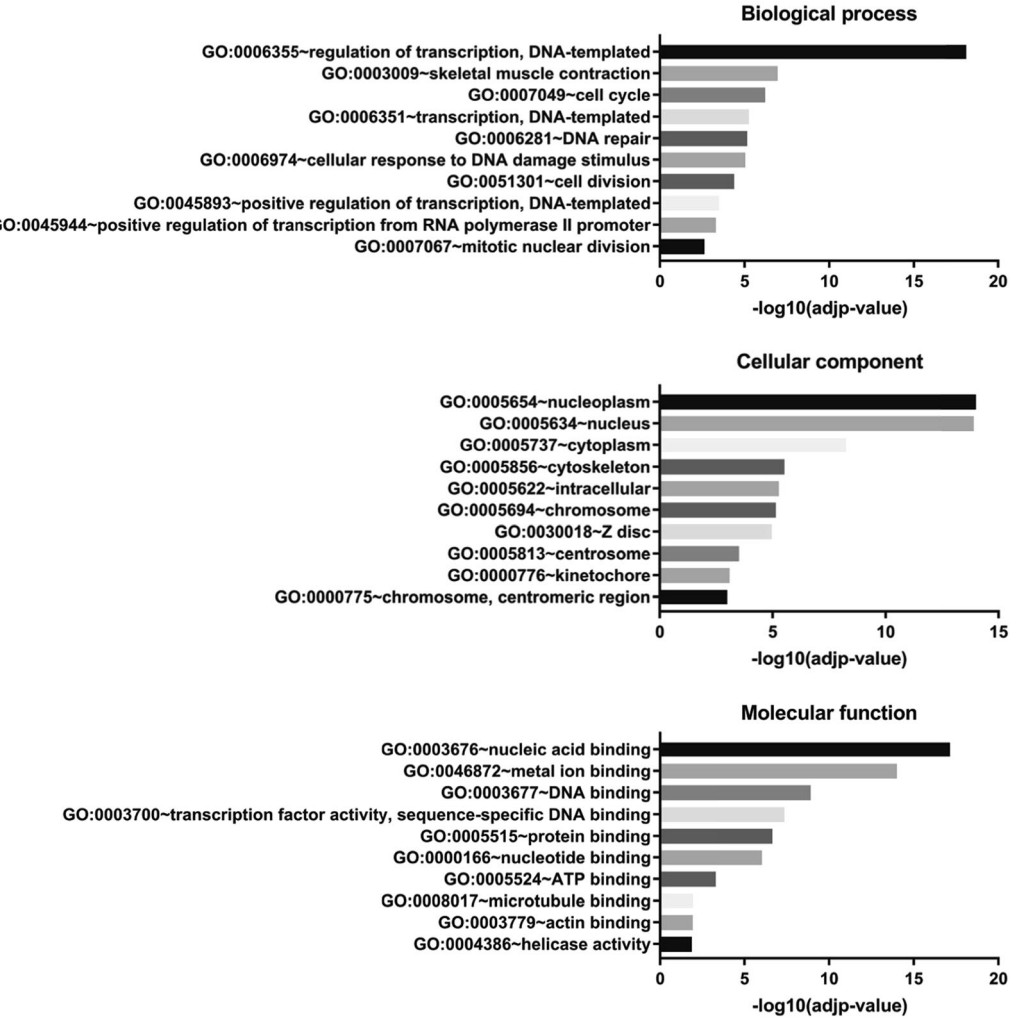

**Fig. 2 Transcriptome profiling reveals changes in gene expression programs between NM1 WT and KO cells.** Differentially expressed genes by at least twofold between WT and KO cells in RNA-Seq analysis were subjected to GO analysis. The top ten enriched biological process, cellular components, and molecular functions terms are shown.

antibody (Supplementary Fig. 2c, d) confirmed increased DNA damage in the absence of NM1 in comparison with WT.

DNA repair and signaling factors are highly regulated through changes in their expression, localization, or posttranslational modifications such as phosphorylation or acetylation[20]. Therefore, we tested whether in primary WT MEFs, NM1 expression changes in response to DNA damage. We treated WT MEFs cells for 2 h with the DNA damage-causing drug etoposide[21] and prepared protein lysates at different time points after the treatment. Results from immunoblots with anti-γH2AX and anti-NM1 antibodies show that there is an increase in NM1 expression as from 6 h after DNA damage with peaks at 12 h post etoposide treatment (Fig. 3e and Supplementary Fig. 4). In addition, results from high-content phenotypic profiling experiments performed on primary WT MEFs reveal that there is a dose-dependent increase in NM1 expression in cells treated with increasing concentrations of etoposide (Fig. 3f).

To study potential changes in apoptosis, we next subjected NM1 KO cMEFs stained with Annexin V-fluorescein isothiocyanate (FITC) and propidium iodide to flow cytometry. We found that in the absence of NM1 there is a significant increase in number of apoptotic cells compared with WT (Fig. 3g). We next subjected NM1 KO cMEFs with a CTB (CellTiter-Blue®) proliferation assay to study if the rate of cell proliferation is affected

in the absence of NM1. Surprisingly, in spite of having more DNA damage, NM1 KO cells show a significant increase in proliferation in comparison with WT cells (Fig. 3h). This is in agreement with a study in human cancer cells, which showed a higher proliferation rate after Myo1C depletion[22]. To understand this in more detail, we studied the cell cycle distribution of propidium iodide-stained WT and KO cMEFs by flow cytometry (Fig. 3i). Results from these experiments indicate that NM1 KO cells display a significantly lower ratio of G0/G1-phase cells ($P < 0.01$) and a significantly higher ratio of S phase ($P < 0.05$).

As the primary MEFs derived from NM1 KO mouse embryos overall exhibit a similar phenotype as the stable NM1 KO cMEFs, we conclude that the observed increased DNA-damage levels are a specific consequence of NM1 deletion. Despite higher DNA damage, NM1 KO cells do not seem to stop at the G1 checkpoint. Rather, they continue to S-phase, suggesting a general unbalance in the regulation of DDR in these cells. Therefore, we conclude that NM1 may have a role as a positive regulator of the DDR.

**NM1 deletion leads to deregulation of p53 pathway**. To study the potential role of NM1 in DNA damage-induced pathways, we compared and annotated genes related to cell cycle progression, DNA damage signaling, and DNA repair, which are differentially

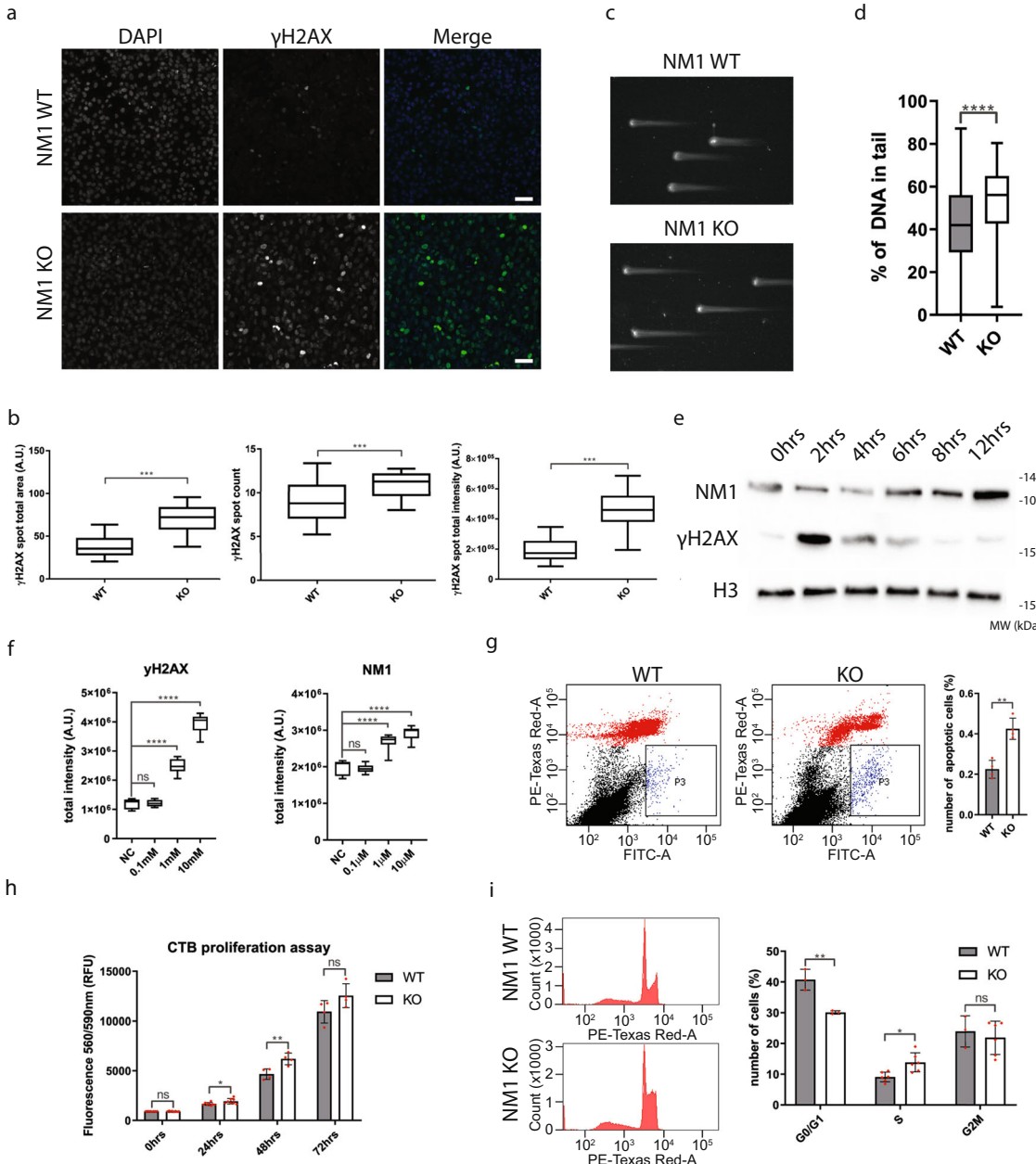

**Fig. 3 NM1 KO cells have more DNA damage and higher proliferation. a** NM1 WT and KO cells stained with γH2AX and DAPI. Scale bar is 20 μm. **b** Quantification of immunofluorescent staining by high-content phenotypic profiling showing an increase in γH2AX foci size, count, and total intensity in NM1 KO cells in comparison with WT cells. Error bars represents minimum and maximum values. $n = 33$ WT, $n = 31$ KO measurements, each containing at least 1000 cells **(c)** Representative comet assay pictures of NM1 WT and KO cells used for analysis. **d** Quantification of DNA content in comets in WT and KO cells. At least 100 comets were measured for each cell type ($n = 106$ WT, $n = 122$ KO). **e** Immunoblots of protein lysates isolated from wild-type cells at different time points after DNA damage stained with antibodies against γH2AX, NM1, and histone H3. **f** Quantification of γH2AX and NM1 staining by high-content screening after increasing the concentration of DNA-damage drug etoposide. Error bars represent minimum and maximum values. $n = 8$ measurements for each condition, each measurement containing at least 500 cells. **g** Representative pictures of flow cytometry analysis of Annexin V-FITC- and propidium iodide-stained NM1 WT and NM1 KO cMEFs cells. Black dots represent live cells, red dots represent necrotic cells, and blue population (P3) represents apoptotic cells. Cells (50,000) plotted for each cell type. The data from $n = 4$ separate flow cytometry experiments are quantified and shown in the graph. **h** CTB proliferation assay on NM1 WT and KO cMEFs. Fluorescence intensity was measured at defined time points after seeding. $n = 4$. **i** Representative histograms of cell cycle distribution of propidium iodide-stained NM1 WT and KO cMEFs cells. $n = 3$. Quantification of the data is shown in the graph. *$p < 0.05$, **$p < 0.01$, ***$p < 0.001$, ****$p < 0.0001$, ns (not significant).

expressed between the KO and WT condition in primary MEFs (Fig. 4a; see Supplementary Table 2 for a full list of genes, with their respective Log2(FC), adjusted *p*-value and distribution within each group). Genes found at the intersection (marked with a star) regulate cell cycle in response to DNA damage and therefore are pooled with other DDR genes. These DDR genes

were manually annotated, divided into subgroups based on their function, and plotted as a heatmap of log2-normalized counts for each WT and KO sample in descending order (Fig. 4b). In agreement with previous data, we found that most of the genes involved in DNA repair and signaling are upregulated in the NM1 KO background (Fig. 4b). For instance, ATM (Ataxia

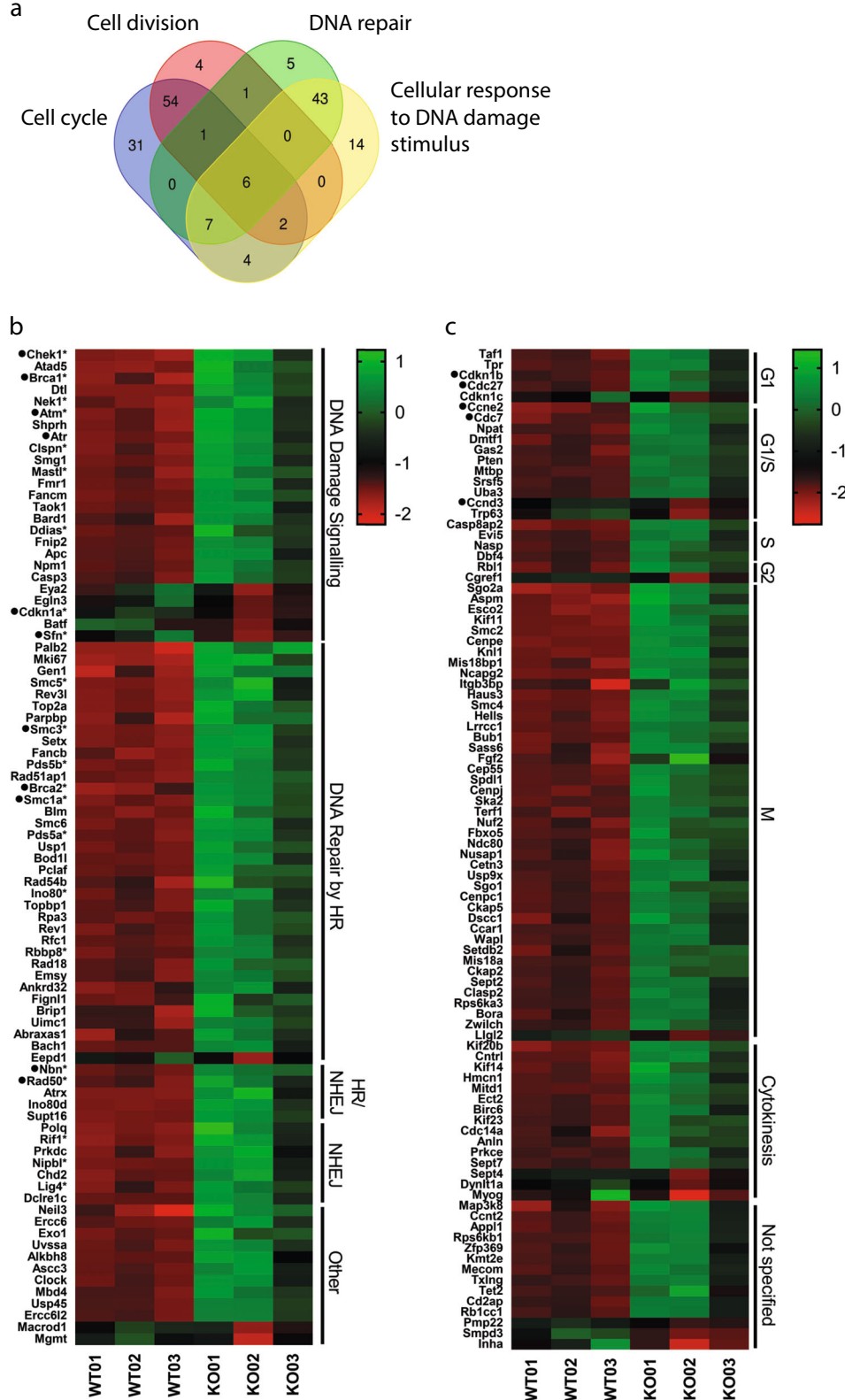

**Fig. 4 Cell cycle and DNA damage response genes are upregulated in NM1 KO cells. a** Venn diagram shows intersection between genes belonging to gene ontology groups "cell cycle," "cell division," "DNA repair," and "cellular response to DNA damage stimulus." **b** Heatmap of genes associated with Gene Ontology terms "cellular response to DNA damage stimulus" and "DNA repair" with at least a twofold change between WT and KO cells. Genes were manually annotated based on the function in DNA damage repair and signaling. Genes marked with * are also present in "cell cycle" or "cell division" gene groups but have primary function in DNA repair-related pathways. **c** Heat map of genes associated with cell cycle and cell division with at least a twofold change between WT and KO cells. Genes were manually annotated according to cell cycle phases and plotted as a heatmap of log2-normalized counts for each WT and KO sample in descending order. Genes marked with ● are discussed in the paper.

Telangectasia Mutated) and ATR (Ataxia Telangectasia and Rad3 Related) kinases, which are the first to recognize and signal DNA damage, are upregulated in NM1 KO cells, as well as in their downstream targets—Chek1 and Brca1. Importantly, ATM and ATR kinase activation leads to a signaling cascade through p53 protein. After DNA damage, p53 is phosphorylated by several kinases and binds to the promoter regions of its target genes to activate their expression. Among these target genes, p21 (*Cdkn1A*) is required at the G1 checkpoint, where it inhibits the activity of Cyclin E and Cyclin A/CDK2 required for the G1/S phase transition[16]. The amount of p21 is p53-dependent and increases after DNA damage[23]. Surprisingly, in spite of increased DNA damage in cells lacking NM1 and overexpression of upstream ATM and ATR signaling kinases, p21 expression is downregulated in the NM1 KO condition. This suggests a potential role for NM1 in regulating p53-dependent p21 expression and, potentially, other p53 target genes. In agreement with this, we evaluated all proven p53 target genes[24] and found enrichment of NM1 at their respective TSS (Supplementary Fig. 3e). An example is provided by the DNA-damage signaling protein, 14–3–3-σ (*Sfn*), which is a direct target of p53[25], and is also downregulated in NM1 KO cells.

Besides the DNA damage signaling, many DNA repair genes are upregulated in KO cells as well. Among others, members of the MRN complex (Mre11, Rad50, and Nibrin; Rpa2 protein; Brca2) or members of the Cohesin complex (Smc1A and Smc3) are all upregulated in KO cells.

Taken together, these results suggest that DNA damage in NM1 KO cells is properly recognized and cells actively repair DNA lesions. However, DNA damage signaling downstream of p53 protein is affected in KO cells, as p21, the main regulator of G1 checkpoint, is downregulated in NM1 KO cells despite increased DNA damage.

**Expression of cell cycle genes in NM1 KO cells is compatible with p21 mutant phenotypes.** Previous reports showed that MEFs with a defect in p21 fail to arrest in G1 phase in response to DNA damage[26]. This scenario is similar to the one depicted in the absence of NM1 in which cells proliferate at a faster pace, there is a reduction in G1 phase and increase in S phase in NM1 KO cells, and, finally, there is a defect in p21 expression.

To start testing this hypothesis, we analyzed the expression of all cell cycle and cell division genes found in the GO analysis (Fig. 2). Genes were manually annotated according to cell cycle phases and plotted as a heatmap of log2-normalized counts for each WT and KO sample in descending order (Fig. 4c). Similar to DDR, most of the cell cycle genes are upregulated in NM1 KO cells. In G1/S phase, where p21 plays a main role, NM1 deletion leads to upregulation of several crucial proteins such as Cyclin E2 (Ccne2), Cdk2, Cdc27, or Cdc7, important for transition from G1 to S phase and subsequent DNA synthesis[27] (Fig. 4c and Supplementary Table 3). Cyclin E forms a complex with Cdk2 and p21 protein expression is inversely proportional to Cyclin E2 and Cdk2 expressions[28], as p21 negatively regulates Cyclin E2 expression by binding to its promoter region[29]. Similarly, in S-phase p21 negatively regulates PCNA[30], which is overexpressed in NM1 KO cells. Most of the differentially expressed genes between KO and WT conditions are cell cycle genes related to mitosis and cytokinesis (Fig. 4c). These genes are upregulated in the KO cells and this is consistent with a role of p21 in G2-phase checkpoint arrest[31,32] and mitosis[33,34], as p21 depletion leads to mitotic defects[34], whereas upregulation of p21 provokes rapid downregulation of genes involved in the execution and control of mitosis[29].

It is also likely that in the absence of NM1, cells try to compensate for the loss of p21 by other mechanisms, possibly by

overexpressing p27 (*Cdkn1b*), a p21-related Cdk-inhibitor, which is upregulated, whereas its direct targets Cdk4 and Cyclin D3 (*Ccnd3*) are downregulated in NM1 KO cells.

In conclusion, transcriptional profiling, cell cycle progression, and increased proliferation in NM1 KO cells resemble phenotypes described in the p21-depleted cells. We therefore propose that NM1 specifically regulates p21, and that the observed phenotypes in NM1 KO cells are mainly the result of insufficient response of cells to endogenous DNA damage caused by a decreased expression of p21. In fact, NM1-dependent p21 downregulation may lead to impairment of cell cycle checkpoints, allowing cells to progress through the cell cycle without repairing DNA and accumulation of DNA damage in the next generations.

**NM1 is required for activation of *p21* gene upon DNA damage.** As both NM1 and p53 interact with the HAT PCAF[2,5,35,36], we next examined whether they are part of the same complex and synergize under DNA damage conditions to activate the *p21* gene. For this, we treated cells with 10 μM Etoposide for 2 h, followed by 10 h incubation in full Dulbecco's modified Eagle's medium (DMEM). We next prepared lysates and subjected them to co-immunoprecipitations with antibodies against NM1, p53, PCAF, and nonspecific rabbit immunoglobulins (IgG) (Fig. 5a and Supplementary Fig. 4). The results show that upon DNA damage, NM1, p53, and PCAF can be co-precipitated from total lysates, whereas control GAPDH stays in flow-through fraction and control IgG does not precipitate any of the proteins. In untreated cells, we found that NM1 and PCAF co-precipitate each other, whereas binding of p53 is considerably reduced (Supplementary Fig. 2e). This indicates that upon DNA damage, p53, NM1, and PCAF are likely to be part of the same complex; however, we cannot say whether the interaction is direct or not.

Next, we set out to investigate the potential involvement of NM1 in *p21* transcription activation upon DNA damage. First, we mapped NM1 ChIP-seq reads along *p21* gene using UCSC genome browser (Supplementary Fig. 3f). As can be seen, NM1 is specifically enriched at the *p21* TSS and its occupancy peak correlates with active histone marks H3K4me3 and H3K9ac (Supplementary Fig. 3f). We therefore examined whether NM1 depletion has an effect on p21 expression in cells. As NM1 KO cells has initially more DNA damage and less p21 protein in comparison with WT cells, they cannot be used as a proper control in DNA damage-dependent ChIP experiments. Therefore, we designed a loss-of-function model to study the immediate effect of NM1 depletion in DDR. For this we applied RNA interference (RNAi) duplexes against the target sequence 5′-GCACACGGCUUGGCACAGA-3′ in the mouse Myo1C gene (siRNA) or control scrambled versions (scrRNA) to MEFs[2]. Immunofluorescence staining with anti-NM1 antibodies show differences in NM1 expression between control (scrRNA) and KD (siRNA) MEFs (Supplementary Fig. 2f). Real-time qPCR and immunoblottings on control and KD MEFs (Supplementary Fig. 2g, h) show ~80% reduction in NM1 mRNA levels and 50% reduction at the protein level 48 h after transfection.

We subjected the NM1-silenced cells to etoposide-induced DNA damage. Analysis of transcription levels was performed by qPCR on total RNA isolated from the cells at different time points following treatment with etoposide using primers specific for the *p21* gene. Control cells transfected with scrambled scrRNA show rapid increase in p21 mRNA levels upon DNA damage (Fig. 5b), which correlates with increasing expression of NM1 protein after etoposide treatment (Fig. 3e and Supplementary Fig. 4). In contrast, NM1 KD cells show significantly decreased p21 expression (Fig. 5b), which is compatible with RNA-Seq data in

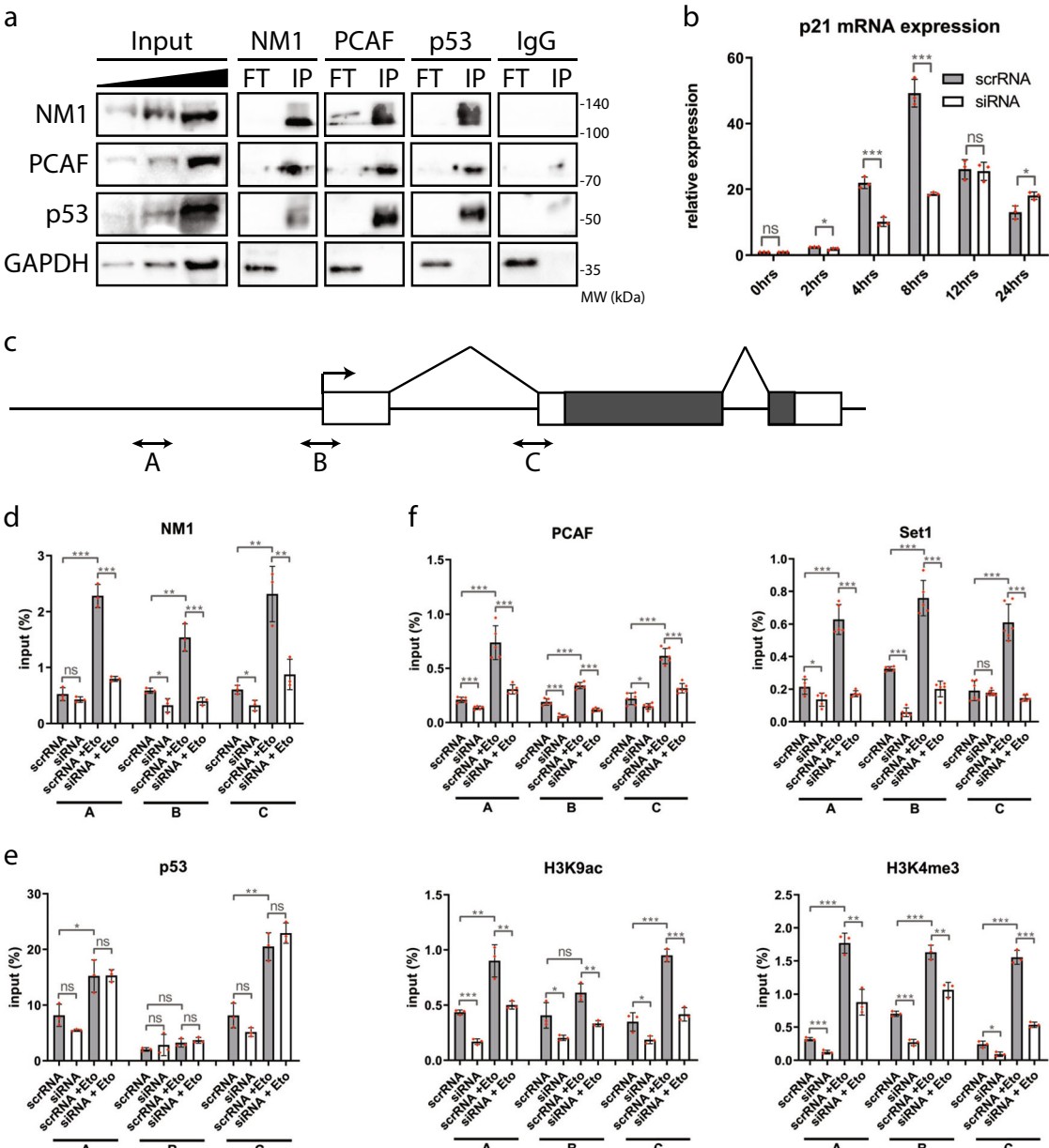

**Fig. 5 NM1 is required for activation of the *p21* gene upon DNA damage. a** Co-immunoprecipitation of proteins with antibodies against NM1, PCAF, p53, or control rabbit IgG and subsequent western blot analysis show that after DNA damage NM1, PCAF, and p53 are immunoprecipitated by each other (IP), whereas GAPDH control is in unbound flow-through fraction (FT). **b** RT-qPCR analysis of p21 expression after DNA damage. Cells transfected with control scrRNA or NM1 siRNA were treated with etoposide for 2 h followed by incubation in normal medium. Total RNA was isolated in each time point and subjected to RT-qPCR analysis with p21- and GAPDH-specific primers. Expression of p21 was defined relatively to expression of GAPDH. $n = $ . **c** Schematic representation of *p21* gene and localization of three primer sets used in the analysis. Boxes represent exons. Grey boxes represent coding region. Arrow represents transcription start site (TSS). Primer pair A is located in the promoter region, Primer pair B is located in TSS, and Primer set C in the putative p53-binding site[37] (Fig. 5c). **d**–**f** NM1 WT (scrRNA) or depleted cells (siRNA) were treated (scrRNA + Eto, siRNA + Eto) or untreated (scrRNA, siRNA) with etoposide and used for chromatin immunoprecipitation with NM1 (**d**), p53 (**e**), or PCAF, Set1, H3K9ac, and H3K4me3 (**f**), and subsequent qPCR analysis with three sets of primers covering *p21* gene. $n = 3$, *$p < 0.05$, **$p < 0.01$, ***$p < 0.001$, ns (not significant).

NM1 KO cells. Seeing that NM1 occupies the TSS of *p21* (Supplementary Fig. 3f) and regulates p21 mRNA synthesis (Fig. 5b), we next examined whether NM1 depletion affects *p21* transcription by interfering with p53 binding and/or by inducing changes in epigenetic marks at the *p21* promoter upon DNA damage. For this, we performed ChIP with anti-NM1 and anti-p53 antibodies on chromatin isolated from control WT and siRNA-mediated NM1 KD MEFs, treated or untreated with etoposide (Fig. 5c–e). The immunoprecipitated chromatin was

then analyzed by RT-qPCR using three sets of primers designed to target different regions of the *p21* gene containing a promoter, TSS, and first intron carrying possible p53-binding site[37] (Fig. 5c). Using all sets of primers, in control cells we found increased levels of NM1 occupancy upon DNA damage (scrRNA vs. scrRNA + Eto) at all three *p21* gene regions under analysis, whereas p53 occupancy increased at promoter and intronic regions but not at TSS (Fig. 5d, e). NM1 depletion did not have significant effects on p53 binding before and after DNA damage (Fig. 5e). These results

suggest that NM1 is recruited to the *p21* (*Cdkn1A*) gene upon DNA damage but it is not required for p53 binding. Next, we performed ChIP analysis using antibodies against PCAF, Set1, H3K9ac, and H3K4me3 (Fig. 5f). As above, these experiments were carried with or without etoposide treatment on cells subjected to NM1 depletion by RNAi. Results from the reverse transcriptase (RT)-qPCR analysis with the same *p21*-specific primers revealed a specific drop in both PCAF and Set1 levels in the NM1 KD cells, and only marginal increment in occupancy levels upon etoposide treatment (Fig. 5f). Compatible with these findings, H3K9ac and H3K4me3 levels were comparably reduced in the absence of NM1 (Fig. 5f).

Altogether, these experiments suggest that NM1 binds to *p21* promoter and binding is enhanced upon DNA damage. Once recruited to the promoter, NM1 appears to interact with p53 and, importantly, NM1 facilitates recruitment of both PCAF and Set1, which in turn leads to acetylation of H3K9 and methylation of H3K4, to activate *p21* gene transcription.

## Discussion

Exogenous and endogenous DNA damage that cause DSBs can lead to genomic instability or cell death[38]. To repair DSBs, one of the mechanisms the cell employs is undergoing cell cycle arrest through activation of the p53 signaling pathway[39]. Here we identify NM1 as a potential regulator of p53 signaling. We report that upon DNA damage, NM1 is recruited to the regulatory region of the *p21* gene and controls its p53-dependent expression through a chromatin-based mechanism. Overall, we show that NM1 facilitates deposition of the HAT PCAF and the HMT Set1 to the TSS for histone acetylation and methylation, to trigger *p21* gene transcription (Fig. 6). These observations suggest a role for NM1 in the transcriptional response to DNA damage and its potential involvement in genome stability.

NM1 has been described as a general transcription factor binding across the mammalian genome, occupying both transcribed and non-transcribed regions[2,40]. At TSS, NM1 regulates transcription by primarily controlling the epigenetic state of chromatin[2]. Here, results from high-content phenotypic profiling show that NM1 depletion globally affects active (H3K9ac, H3K27ac, and H3K4me3) and repressive (H3K9me3) epigenetic marks, and these changes in the epigenetic landscape leads to transcriptional reprogramming of NM1 KO cells.

Chromatin alterations affect the overall organization of the genome and have impact on cellular functions. In this context, cytoskeletal proteins are beginning to emerge as key factors, regulating the state of chromatin and affecting organization and stability of the genome[1]. For instance, in MEFs lacking β-actin, heterochromatin is redistributed, active and repressive epigenetic marks are altered, cells are transcriptionally reprogrammed, and are not efficiently transdifferentiated to neurons[41,42]. In the present study, loss of NM1 similarly induces epigenetic changes that affect global transcription and specifically decrease the ability of cells to respond to DNA damage. Strikingly, we found that primary NM1 KO MEFs show constitutive γH2AX-positive foci and increased proliferation rates. Compatible with elevated number of γH2AX-positive foci, GO analysis on differentially expressed genes between NM1 KO and WT cells revealed a bias towards genes that are either directly or indirectly involved as gatekeepers of genome stability in response to DNA damage. Cell cycle genes and genes involved in DNA repair are primarily upregulated in the absence of NM1, suggesting that under these conditions the cell may progress through the cell cycle at a faster pace and cell cycle arrest checkpoints normally activated in response to DNA damage are overruled. Importantly, these phenotypes were observed using three different NM1 loss-of-function systems (primary MEFs derived from NM1 KO mice prepared by Cre/Lox technology, stable NM1 KO MEFs prepared by Crispr/Cas technology, and NM1 depleted MEFs prepared by siRNA technology), supporting a highly specific function of NM1 in the transcriptional response to DNA damage.

Based on the above, a prediction is that p53 signaling is negatively regulated and this leads to an accumulation of DNA lesions. In a mammalian cell, an average 70,000 DNA lesions/day are normally repaired by different mechanisms[43]. Remarkably, in NM1 KO cells, most of the DNA repair genes relating to homologous recombination and non-homologous end joining are upregulated. In contrast, we found that NM1 deletion leads to p21 downregulation and accumulation of DNA lesions as revealed by increased numbers of γH2AX foci and longer comet tails in a single-cell electrophoresis experiment. From a mechanistic point of view, these DNA lesions may accumulate because of decreased chromosome territory rearrangement upon DNA damage. NM1 binds directly to chromatin[2] and together with other myosin species it has been suggested to facilitate chromosomal rearrangements and long directed movement of chromosomal loci to nuclear regions more favorable for DNA repair[12,13]. Therefore, NM1-dependent loss of a three-dimensional genome reorganization may negatively regulate transcription by changing the architecture of the genome. However, our findings suggest that in response to DNA damage NM1 also serves an important function at the gene level. We propose that the *p21* (*Cdkn1A*) gene is one of the primary targets whose activity is dysregulated in NM1 KOs and this, in turn, leads to loss of integration of DNA repair mechanisms and cell cycle arrest. Upon DNA damage, NM1 binds to both the promoter and TSS of the *p21* gene, and is necessary for deposition of the active epigenetic marks H3K9ac and H3K4me3. p53 binds to nucleosomes within the *p21* promoter, leading to nucleosome loss, and regulates transcription by regulating Pol II DNA-binding functions[44,45]. This suggests that at the *p21* promoter there is no pre-requirement for NM1 to open the chromatin and facilitate p53 binding itself, although we cannot exclude that NM1 may facilitate p53 posttranslational modifications. Seeing that NM1 and p53 can be co-precipitated, we speculate that upon DNA damage NM1 and p53 enhance *p21* transcription by bringing promoter and TSS in close proximity. This would then lead to NM1-dependent recruitment of both PCAF and Set1 for deposition of active epigenetic marks at TSS, thus contributing to *p21* transcription activation. We speculate that this mechanism ensures that p53 maintains a poised polymerase at the gene promoter ready for a rapid response to NM1-dependent transcription activation upon DNA damage.

In conclusion, we propose that at the *p21* TSS, NM1-dependent deposition of epigenetic marks could be a unique epigenetic strategy to rapidly activate poised p53 target genes and maintain their expression in response to DNA damage.

## Methods

**Cell culture, reagents, and antibodies**. All cell lines were cultured in DMEM medium with 10% fetal bovine serum, 100 U/ml penicillin and 100 mg/ml streptomycin (Millipore-Sigma) in a humidified incubator with 5% $CO_2$ at 37 °C.

Initial experiments were performed on primary MEFs derived from 13.5 days NM1 WT and KO embryos[46]. In other experiments were used immortalized MEFs (ATCC® CRL-2752) or cell lines derived from these immortalized cells.

The antibodies against H3K9me3 (ab8898), H3K27ac (ab4729), H3K4me1 (ab8895), H3K4me3 (ab8580), H3K9ac (ab10812), H3 (ab1791), γH2AX (ab2893), p53-K370 (ab183544), PCAF (ab12188), Set1 (ab70378), GAPDH (ab8245), and nonspecific rabbit IgG isotype control (ab37415) were purchased from Abcam (Cambridge, MA, USA). The antibody against NM1 has been previously characterized[2]. Alexa Fluor 555 Goat Anti-Rabbit (ab150078) and Alexa Fluor 488 Goat Anti-Mouse (ab150117) were used as secondary antibodies for immunofluorescence, and

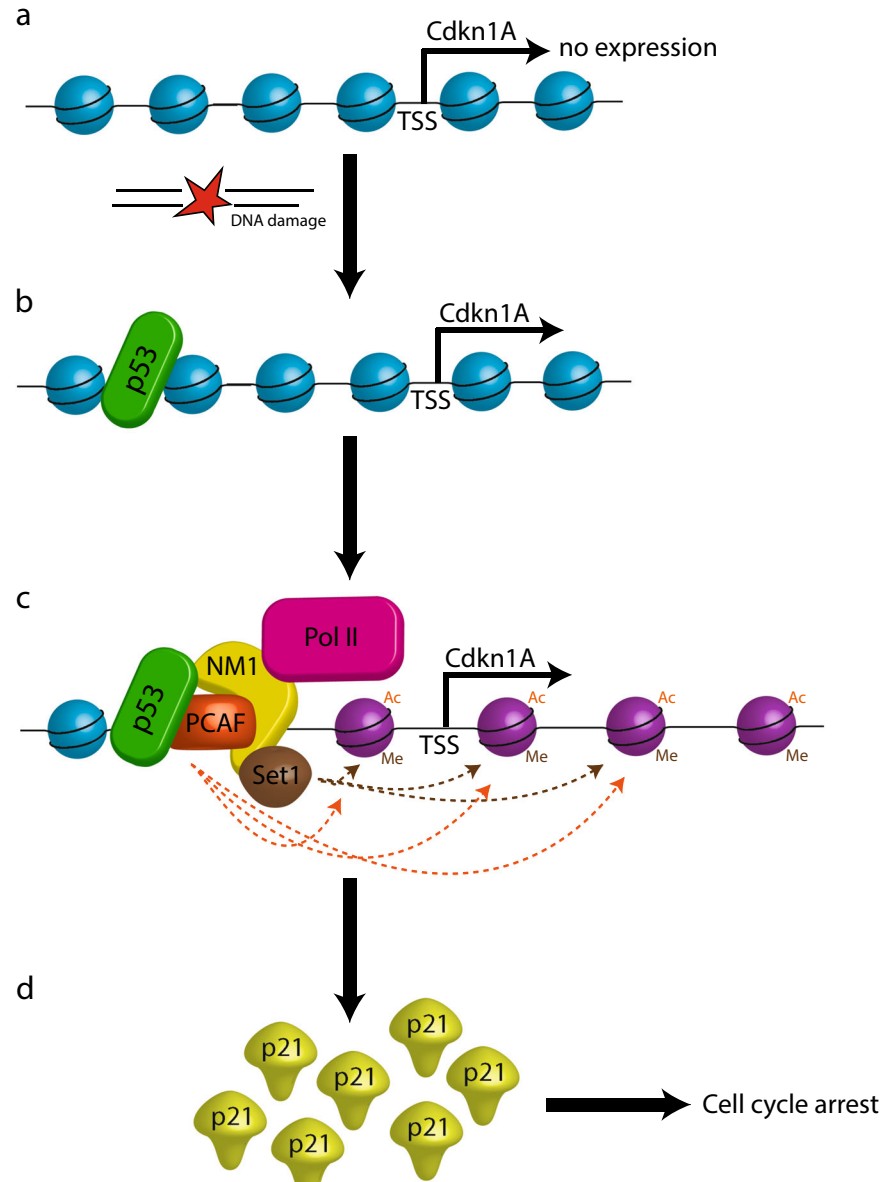

**Fig. 6 A speculative model summarizing the potential role of NM1 in the transcriptional response to DNA damage. a** Under normal conditions, chromatin around *p21* is packed and unfavorable for transcription. **b** After DNA damage, p53 is activated and binds to its responsive elements within *p21* gene. **c** NM1 is stabilized on the *p21* promoter by p53 and brings HAT PCAF and HMT Set1 to acetylate and methylate surrounding histones. This leads to opening of chromatin, allowing RNA Polymerase II to transcribe the *p21* gene. **d** Newly made p21 arrest cells in G1/S phase of cell cycle till the DNA damage is repaired.

horseradish peroxidase (HRP)-fused Goat Anti-Rabbit (ab6721) and Rabbit Anti-Mouse (ab6728) secondary antibodies for western blottings were purchased from Abcam. Hoechst 43222 (H1399) and ProLong Gold Antifade Mountant with 4′,6-diamidino-2-phenylindole (DAPI; P36931) were purchased from Invitrogen, Waltham, MA, USA. All antibodies have been used according to the manufacturers' protocols.

**NM1 knockout by Crispr/Cas9.** For preparation of stable NM1 KO cells were used immortalized MEFs (ATCC® CRL-2752) and Edit-R CRISPR-Cas9 Gene Knockout system from Dharmacon, Lafayette, CO, USA, and GE Healthcare, according to the manufacturers' protocol. Shortly, synthetic guide crRNA against NM1-specific N terminus (sequence: 5′-CCGGCAGGAUGCGGCUACCGU UUUAGAGCUAUGCUGUUUUG-3′) was co-transfected with tracrRNA and Cas9 plasmid by using DharmaFECT Duo reagent. Twenty-four hours after transfection, puromycin selection was used to discard untransfected cells. After 2 weeks, single colonies were selected and tested for mutations by T7 endonuclease I cleavage assay (NEB), Sanger sequencing, and western blotting. NM1 KO cell clones were propagated and stored for future use. We named these stable cells as NM1 KO cMEFs.

**NM1 KD by RNAi.** Immortalized MEFs (ATCC® CRL-2752) were seeded 1 day before the transfection for 40–60% confluence. RNAi duplexes against the target sequence 5′-GCACACGGCUUGGCACAGA-3′ in the mouse *Myo1C* gene (NM1 RNAi) or control scrambled versions (scrRNAi) were purchased from Dharmacon, Lafayette, CO, USA, and GE Healthcare. Cells were transfected with Lipofectamine RNAiMax (Invitrogen, Waltham, MA, USA) at a final concentration of 30 nM according to the manufacturer's protocol. The subsequent experiments and analysis were performed 48–72 h after transfection.

**Neutral comet assay.** Cells were grown until 80% confluency, tripsinized, and resuspended in cold phosphate-buffered saline (PBS). 20 µl of cells were mixed with Low Melting Point agarose (0.5%) and spread over a pre-coated microscope slide (0.3% Normal Melting Point agarose). Once the agarose solidified, slides were incubated in chilled lysis buffer (2.5 M NaCl, 100 mM EDTA, 10 mM Trizma base, 1% Triton X-100, NaOH pH 10) for 1 h at 4 °C and in 1× TAE (Tris-Acetate-EDTA) buffer for 30 min at 4 °C. The electrophoresis was run at 35 V for 15 min. Finally, the slides were washed in chilled distilled water and stained with DAPI (BIOTUM Ltd, 10 µg/ml). Samples were visualized using a Widefield Leica

DMI6000 microscope (Leica Microsystems) and the percentage of DNA in the tail of the comets was quantified using the software CometScore 2.0. At least 100 comets per sample were measured in three separate, independent experiments.

**Cell proliferation assay**. NM1 WT and KO cMEFs cells were seeded in 96-well plates to 20% confluence. The cell proliferation was measured by CellTiter-Blue® Cell Viability assay (Promega) according to the manufacturer's protocol. On a given time, 20 µl of CTB reagent was added to each well and incubated at 37 °C for 2 h, to allow resazurin to reduce to fluorescent resorufin. Cell proliferation was quantified by measuring total fluorescence at 560ex/590em nm using Synergy H1 microplate reader (BioTek). Eight replicates were used for each condition and each time point.

**Analysis of cell cycle and apoptosis by flow cytometry**. For cell cycle experiments, WT and KO cMEFs were grown for 24 h for 50–70% confluence. Cells were trypsinized, washed with ice-cold 1× PBS, and fixed by slow addition of 70% ethanol. After 2 h incubation at 4 °C in rotation, cells were spin down, washed with 1× PBS, and treated with RNAse A and propidium iodide for 30 min at 37 °C.

For apoptosis assay, we used Annexin V-FITC Apoptosis Staining/Detection Kit (Abcam, ab14085) according to the manufacturer's protocol. Overnight grown cells were trypsinized and collected by centrifugation. Cells were resuspended in 1× binding buffer with Annexin V-FITC and propidium iodide, and incubated for 5 min in dark before analysis.

Analysis was performed by flow cytometer BD FACSAria II (BD Bioscience, San Jose, CA). Three replicates were used for the analysis for each sample, each containing 100,000 cells.

**DNA-damage drug treatment**. MEFs were grown for 24 h for 70–80% confluence. In most of the experiments, etoposide was diluted to 10 µM in DMEM medium, added to the cells, and incubated for 2 h. Then the medium was changed back to normal DMEM medium for 10 h before subsequent analysis. For drug-dependent experiments, different etoposide concentrations have been used—0.1 µM (Eto 1×), 1 µM (Eto 10×), and 10 µM (Eto 100×). For time-lapse experiments, cells were incubated with 10 µM etoposide in DMEM medium for 2 h, followed by incubation in normal DMEM medium for designated periods of time.

**Immunofluorescence**. Cells grown on glass coverslips in a 24-well plate were fixed with 4% paraformaldehyde for 20 min and then permeabilized using 0.5% Triton X-100 for 10 min. Cells were stained with primary antibodies against NM1 and γH2AX for 2 h. The cells were washed with PBS with 0.5% Tween-20 (PBST) buffer and stained with respective secondary antibodies for 1 h, and mounted with Pro-long containing DAPI. The staining was observed using Leica DMI6000 Fluorescence Microscope (Leica, Germany). Analysis of images was done using ImageJ software.

**High-content phenotypic profiling**. Cells were cultured in 364-well clear-bottom assay plate (Corning, Corning, NY, USA) at a density of 2500 cells/well. After fixation in 4% formaldehyde for 10 min, cells were permeabilized with 0.5% Triton X-100 for 10 min. The cells were stained with primary antibodies overnight followed by three washes with PBST buffer. The cells were stained with secondary antibodies for 1 h and with Hoechst (5000× dilution) for 20 min. The cells were washed twice with PBST and stored in PBS buffer. The plate was analyzed via the Cellomics ArrayScan XTI High-Content Screening platform (Thermo Fisher Scientific).

The image analysis was performed using the Compartment Analysis BioApplication software (Thermo Fisher Scientific). Primary objects were defined by the Hoechst stained nuclei. Fluorescent measurements were quantified per well. In each experiment, at least three wells per condition, each containing at least 5000 cells, were used. Each experiment was repeated three times with similar results.

**RNA-Seq library preparation, sequencing, and analysis**. Total RNA was extracted from three replicates of NM1 WT and NM1 KO primary MEFs with TRI Reagent (Millipore-Sigma) according to the manufacturer's protocol. The RNA-Seq library was prepared by using the TruSeq Stranded mRNA Library Prep Kit (Illumina) and sequenced with the HiSEq 2500 sequencing platform (performed at the NYUAD Sequencing Center). All of the subsequent analysis, including quality trimming, was executed using the BioSAILs workflow execution system. The raw reads were quality trimmed using Trimmomatic (version 0.36) to trim low-quality bases, systematic base calling errors, as well sequencing adapter contamination. FastQC was used to assess the quality of the sequenced reads pre/post quality trimming. Only the reads that passed quality trimming in pairs were retained for downstream analysis. The quality-trimmed RNA-Seq reads were aligned to the *Mus musculus* GRCm38 (mm10) genome using HISAT2 (version 2.0.4). The resulting SAM alignment files for each sequenced sample were then converted to BAM format and sorted by coordinate using SAMtools (version 0.1.19). The BAM alignment files were processed using HTseq-count, using the reference annotation file to produce raw counts for each sample. The raw counts were then analyzed using the online analysis portal NASQAR (http://nasqar.abudhabi.nyu.edu/), to

merge, normalize, and identify differentially expressed genes. Differentially expressed genes by at least twofold (log2(FC) ≥ 1 and adjusted *p*-value of <0.05 for upregulated genes, and log2(FC) ≤ −1 and adjusted *p*-value of <0.05 for down-regulated genes) between the NM1 WT and KO MEFs were subjected to GO enrichment using DAVID Bioinformatics (https://david.ncifcrf.gov/). Venn diagram was produced by Bioinformatics and Evolutionary Genomics platform (http://bioinformatics.psb.ugent.be/webtools/Venn/). RNA-Seq data on NM1 KO and WT primary MEFs were deposited in the Gene Expression Omnibus (GEO) database under accession number GSE133506.

**ChIP-seq analysis**. NM1 ChIP-seq data set was previously published[2] and raw data stored in GEO database (accession number GSE66542). The raw reads were quality processed as previously stated. The quality-trimmed reads were then aligned against the mouse reference genome (GRCm38.p4) using BWA-MEM. The resulting BAM alignments were then processed through PICARD tools, to clean, sort, and deduplicate (PCR and Optical duplicates). The processed alignments were analyzed with DeepTools2 (version 2.5.1) and resulting BigWig files were then passed to computeMatrix, which generates a matrix of scores per genomic region that is needed by the plotHeatmap tool. Finally, density maps of ChIP-seq reads for NM1 ±5 kb from the TSS of selected genes were produced by plotHeatmap.

**Western blotting**. Total lysate of MEFs were collected in RIPA buffer (50 mM Tris-HCl pH 7.5, 150 mM NaCl, 1 mM EDTA, 1% NP-40, 0.5% sodium deoxycholate, 0.1% SDS) with protease inhibitors, using the Pierce BCA (12-Bicinchonic acid) protein assay kit (Thermo Fisher Scientific) was performed on samples and 20 µg of protein per sample was loaded to the 10% SDS-polyacrylamide gel electrophoresis (PAGE) gel. The extracts were separated under reducing conditions and transferred to polyvinylidene difluoride membrane. After staining with primary antibodies and subsequent washes in 1× TBST buffer (20 mM Tris, 150 mM NaCl, 0.1% Tween-20), immunoblots were stained with HRP-fused secondary antibodies. Protein bands were developed with ECL Western Blot Substrate (Bio-Rad) and imaged by a ChemiDoc MP Imaging system (Bio-Rad). The quantification of the blots was performed by ImageJ software and at least three blots were used for analysis for each samples.

**Co-immunoprecipitations**. Etoposide-treated and -untreated cells were lysed with NP-40 buffer (20 mM Tris-HCL pH 8, 137 mM NaCl, 10% glycerol, 1% Nonidet P-40, 2 mM EDTA, freshly added protease inhibitors) and agitated for 30 min at 4 °C. After centrifugation, supernatant protein concentration was measured by BCA assay and 10% of supernatant stored as an input. Protein lysate (2 mg) was mixed with 10 µg of NM1, p53-K370, PCAF, or unspecific rabbit IgG antibodies, respectively, and incubated overnight at 4 °C under agitation. Subsequently, 250 µg of pre-washed protein A/G magnetic beads (Pierce) was added to each sample incubated at 4 °C under agitation for 4 h. Protein beads bound to protein complexes were collected with a magnetic stand and flow-through supernatant fraction saved for the analysis. Beads were washed with TBST buffer containing 0.1 M NaCl and protein complexes eluted directly to SDS-PAGE Laemeli reducing sample buffer.

**Quantitative RT-PCR**. Total RNA was extracted using RNazol (Sigma) according to the manufacturer's instructions. RNA (500 ng) per sample was reverse transcribed using RevertAID First Strand cDNA Synthesis Kit (Thermo Fisher Scientific) and cDNA was cleaned from residual gDNA by Turbo DNA-free kit (Ambion). Diluted cDNA was subjected to quantitative real-time PCR analysis using Maxima SYBR Green qPCR Mix (Thermo Fisher Scientific) and the three-step cycling protocol on the Real-Time Thermal Cycler (Thermo Fisher Scientific). Relevant primers were designed against the selected genes (Supplementary Table S4). qPCR analysis was performed in triplicates and in three different experiments. The expression data were analyzed by normalizing each sample to the GAPDH expression.

**ChIP and qPCR analysis**. ChIP analysis was performed on cells transfected with siRNA or scrRNA, and treated or untreated with etoposide. Approximately 20 million of cells were used for each ChIP. Cells were grown to 40% confluence and transfected as described above. After 36 h, cells were treated with 10 µM etoposide in DMEM media for 2 h, followed by incubation in normal media for additional 10 h. Cells were fixed with 1% formaldehyde 48 h post transfection and the reaction was stopped with 0.125 M glycine. Cells were then lysed with the lysis buffer (10 mM Tris pH 8.0, 10 mM NaCl, 0.2% NP-40) supplemented with protease inhibitors. The nuclei were recovered and resuspended in nuclei lysis buffer (50 mM Tris pH 8.1, 10 mM EDTA pH 8.0, 1% SDS, ddH2O) with protease inhibitors. Chromatin was sonicated by Covaris E220 instrument using two rounds of DNA shearing (peak incident power 175, duty factor 10%, 200 cycles per burst, 180 s treatment time). Final DNA fragment size was checked by DNA electrophoresis to be 200–500 bp. Sheared chromatin was divided equally for immunoprecipitation with antibodies fused to Dynabeads (Thermo Fisher Scientific) in IP Dilution buffer (20 mM Tris pH 8.1, 2 mM EDTA pH 8.0, 150 mM NaCl, 1% Triton X-100, 0.01% SDS, ddH2O). 10 µg anti-NM1, 5 µg anti-p53, 10 µg anti-PCAF, 5 µg anti-Set1, 5 µg anti-H3k9ac, and 5 µg of anti-H3K4me3

antibodies were used in each ChIP condition. 10% of sheared chromatin served as an input control. Samples were incubated overnight, rotating at 4 °C. The IP samples were washed with IP Wash buffer 1 (20 mM Tris pH 8.1, 2 mM EDTA pH 8.0, 50 mM NaCl, 1% Triton X-100, 0.1% SDS, ddH2O) and IP Wash buffer 2 (20 mM Tris pH 8.1, 1 mM EDTA pH 8.0, 0.25 M LiCl, 1% NP-40, 1% sodium deoxycholate monohydrate, ddH2O). Reverse crosslinking and elution of DNA were performed by using the elution buffer (100 mM NaHCO₃, 1% SDS, ddH2O). The samples were incubated with 5 M NaCl and 10 mg/ml of RNase A at 65 °C for 1 h. Proteinase K (20 mg/ml) was then added and incubated at 55 °C for 2 h. The samples were then placed on a magnet and the immunoprecipitated DNA samples in the supernatant along with the input samples were purified using ChIP Purification Kit (Zymo Research), according to manufacturer's protocol. They were diluted in 12 µl of elution buffer and the concentration was measured by Qubit (Thermo Fisher Scientific).

qPCR analysis was performed in triplicates, containing 3 µl of 30× diluted immunoprecipitated sample or 150× diluted input sample in each reaction mixed with Maxima SYBR Green/Rox qPCR Master mix (Thermo Fisher Scientific) and appropriate set of primers (Supplementary Table S4), followed by three-step cycling protocol in the Real-Time Thermal Cycler (Thermo Fisher Scientific). The data were analyzed by normalizing each sample to the adjusted input.

**Statistics and reproducibility**. GraphPad Prism 8.3.0 software was used for statistical analysis. Unpaired $t$-test was used in all the experiments, unless otherwise stated in text. Error bars in boxplots represents minimum and maximum values. Error bars in bar charts represents SD. Statistical significance is marked with $*p < 0.05$, $**p < 0.01$, $***p < 0.001$, $****p < 0.0001$, ns (not significant). Each experiment was performed at least in triplicates. Exact description of sample sizes and number of replicates is defined for each experimental procedure in Methods.

**Reporting summary**. Further information on research design is available in the Nature Research Reporting Summary linked to this article.

## Data availability

The data sets generated during and/or analyzed during the current study are publicly available in the Gene Expression Omnibus (GEO) database repository at accession number GSE133506 and accession number GSE66542.

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

## Acknowledgements

This work is primarily supported by grants from New York University Abu Dhabi, the Sheikh Hamdan Bin Rashid Al Maktoum Award for Medical Sciences, the Swedish Research Council (Vetenskapsrådet), and the Swedish Cancer Society (Cancerfonden) to P.P. We thank the NYU Abu Dhabi Center for Genomics and Systems Biology, in particular Marc Arnoux and Mehar Sultana for technical help, as well as Core Technology Platform Resources, including the NYU Abu Dhabi imaging center. We appreciate the computational platform provided by NYUAD HPC team and are especially thankful to Yousif Ayman and Nizar Drou for technical help. The study was also supported by the Grant Agency of the Czech Republic (17–09103s, 16–03346s, and 15–08738s). We acknowledge the Microscopy Centre–Light/Electron CF, IMG AS CR supported by the Czech-BioImaging large RI project (LM2015062 funded by MEYS CR) for technical help. The results achieved with institutional support were obtained with the support of long-term conceptual development of the scientific organization (RVO: 68378050). This publication is also supported by the project "BIOCEV–Biotechnology and Biomedicine Centre of the Academy of Sciences and Charles University" (CZ.1.05/1.1.00/02.0109), from the European Regional Development Fund.

## Author contributions

T.V., K.S., S.F., M.E.C., and R.H. performed experiments. T.V. and P.P. generated ChIP-seq and RNA-Seq data sets. T.V., P.H., and P.P. analyzed the data. P.P. supervised the overall study. T.V. and P.P. wrote the manuscript.

## Competing interests

The authors declare no competing interests.
