## [Peer Review File · Communications Biology]

Reviewers' comments:

Reviewer #1 (Remarks to the Author):

This is an interesting manuscript on an emerging field linking myosin to DNA damage signaling.

My concerns are eminently technical and, at the Editor's discretion, need to be addressed for this manuscript to be published in this Journal.

Major points.

My main concern is that, as it reads in the M&M section, immortalized cell lines were used for most experiments. As these likely were expanded over a long time it is possible they drifted toward unexpected phenotypes. It is important to either complement the KO cell line or to back these results up by acute siRNA .

Fig 3A is better supported by higher resolution images and/or immunoblots

As gH2AX is only an indirect measure of DNA damage, other assays (like comet assays) are necessary to demonstrate it

Fig 3E needs normalization with a histone

Fig S2D-E: the observation that siRNA kd does not recapitulate the KO phenotype (gH2AX signals) is a significant concern (see my first point above).

Fig 4B: as some DNA repair genes are cell-cycle regulated, some sort of normalization on different proliferation rates/cell cycle is useful here.

Fig 3F: the observation that KO cells have more DNA damage and, despite that, proliferate more is perplexing and it is not "in line" or expected as the authors write

Fig 5A needs IP in undamaged cells: otherwise the statement that such interaction is stimulated by DNA damage is not warranted

Minor:

Ref 18 is not a strong reference as heavy ions only were used.

Reviewer #2 (Remarks to the Author):

This paper reports on the possible role of nuclear myosin I (NMI) in the DNA damage/ repair response. Since its discovery almost 20 years ago, progress on understanding the roles of NMI in the nucleus have been slower than on nuclear actin. In this respect, this paper is timely and of interest.

My main issue with this paper is that it is descriptive and largely correlative in nature. Many pages in the Results section are devoted to listing possible interacting partners. While the RNA-Seq studies are important, at this stage they do not offer mechanistic insights into the role of NMI. There is an effort to discover mechanistic insights, but those data are not definitive. For instance, the observation that p21 and 14-3-3 are downregulated in NMI KO cells does not mean there is a direct line between NMI

and the expression of these proteins. As pointed out on page 13, this pathway “may lead to impairment of cell cycle arrest” and most of the conclusions are qualified in a similar manner.

There are also concerns regarding the knock out cells and the specificity of the antibody. Fig. 3A shows a blot in which the putative NMI band increases over time following etoposide treatment. Is this blot on KO cells or embryonic fibroblasts? How do these levels of NMI in KO cells compare to those in eto treated wild type cells? This is a little odd because there appears to be significant knock-down of NMI but the cells respond in a robust way to etoposide. Furthermore, if the antibodies cannot distinguish between NMI and cytoplasmic MI, how can one be sure this is truly due to an increase in NMI? One possibility is qPCR.

Regarding the knock outs, were the sequences used to target NMI specific to the 5' region unique to NMI or did they target the coding region that includes MI? The latter would knock down both and complicate the analysis because MI can substitute for NMI. If the 5' region of NMI was targeted, what was done to ensure the absence of off target effects? This region is rich in GC and such regions tend to be non-specific. Lastly, discrepancies in some of the data between cells derived from the knock out mice and the siRNA cells are disconcerting.

Other Points:

1. What statistical analyses were performed to test the significance of the data? Data in some of the figures are labelled statistically significant despite overlapping error bars. How was significance determined?
2. The Abstract states NMI KO cells exhibited increased proliferation. This appears to be based on Fig. 3F, which shows a barely significant different in growth rate at 24 hours only. This is not very convincing.
3. In Fig. 3A and S2D, is each spot a different cell nucleus? Please provide size markers.
4. What cells were used in Fig. 3E? Are these total cell extracts or nuclear extracts?
4. Fig. S2A: The DAPI panel does not seem to correlate with the other 2 panels. How many microns does the size marker represent?

Reviewer #3 (Remarks to the Author):

Venit et al report a role for NM1 in supporting the p53-dependent induction of p21 following DNA damage. Phenotypic profiling and RNA-Seq of WT and KO MEFs suggest NM1 is important for generating histone marks associated with the induction of DNA damage/cell cycle regulator genes, notably p21. Consistent with these data, the authors find that NM1 is required for the recruitment of PCAF and SET1 to the p21 gene in response to etoposide.

This is potentially an interesting paper but lacks critical pieces of data. In addition, the results and discussion were extensively co-mingled. Even some figure legends were cast as discussion points. Significant editing and recasting will improve the presentation. Additional major comments are listed below:

- 1) Since NM1 KO mice are available, it is disappointing that the authors focused in MEFs as opposed to radiosensitive tissues. It may be that the analysis of tissues or a larger cohort of MEFs would strengthen the fairly weak PCA analysis in Figure S1 (WT3 is as far from WTs 1 and 2 as they are from the KO samples).
- 2) The imaging data in Figure 1 should be confirmed by western blot for at least a subset of the

histone marks. In addition an inordinate amount of text (nearly 3 pages!) was used to describe the initial imaging/RNA-Seq/bioinformatics analyses, most all of which was relegated to supplemental figures. The authors should consider placing Figure 1 in the supplemental figures and replacing it with a more informative figure composed of S1A, S1D and 2.

3) The CTB proliferation data are not convincing and far less robust than the Myo1c data referred to by the authors. This shortcoming may result from the inefficient NM1 knockdown described in Figure S2A where only a subset of the cells appear to be transfected. See also point 6.

4) Figure 4. Since the paper is centered on p21, it would be helpful to label the heatmap to identify which genes in these GO terms are regulated by or related to p21. It would also help the reader if the authors would indicate which part(s) of the 4A is analyzed in 4B and 4C, respectively.

5) The co-IP data in Figure 5 do not show a complex; they only show a series of pairwise interactions. Additional separation methods would be required to suggest NM1. PCAF and p53 form a complex. The text on pages 14 and 15 needs to be revised to accordingly.

6) Figure 5. The failure to induce p21 following etoposide treatment suggests that the NM1 deficient cells will fail to arrest. This possibility should be tested by flow analysis. Consequently, the NM1-deficient cells may well show increased apoptosis, which should also be examined. Correspondingly, the PCR analysis in panels B/D/F should be expanded to test whether NM1 loss affects induction of Puma or Bax

Point-by-point responses to the reviewers comments, Manuscript #COMMSBIO-19-1114T

Overall, we would like to thank the reviewers for their constructive comments. Our response to each of the reviewer are in bold and they immediately follow specific comments.

Reviewer #1 (Remarks to the Author)

This is an interesting manuscript on an emerging field linking myosin to DNA damage signaling.

We appreciate the interest of this reviewer in our work.

My concerns are eminently technical and, at the Editor's discretion, need to be addressed for this manuscript to be published in this Journal.

Major points.

My main concern is that, as it reads in the M&M section, immortalized cell lines were used for most experiments. As these likely were expanded over a long time it is possible they drifted toward unexpected phenotypes. It is important to either complement the KO cell line or to back these results up by acute siRNA

We would like to point out that the NM1 KO cells used in the initial study are not immortalized, they are derived directly from the mouse embryos and therefore very close to a primary cell phenotype. To back up some of the observations in the NM1 KO embryonic fibroblasts - namely increased number of gamma H2AX positive foci and enhanced DNA damage - we have generated NM1 KO MEFs by CRISPR (in the revised manuscript these cells are termed NM1 KO cMEFs). As can be seen in the supplementary results added to the revised submission these cells display increased γ H2AX foci as revealed by both IF and immunoblots (Fig S2, A-B and C-D).

Fig 3A is better supported by higher resolution images and/or immunoblots

A high resolution image is now provided in supplemental figure 2B for NM1 KO cMEFs.

As γ H2AX is only an indirect measure of DNA damage, other assays (like comet assays) are necessary to demonstrate it

We have now performed COMET assays on WT and KO MEFs. These new results are included as part of figure 3 (Figure 3C-D). The results show that in the absence of NM1 there is increased comet signal, compatible with a higher degree of DNA damage.

Fig 3E needs normalization with a histone

Thanks for pointing this out. The blot is now included in the figure.

Fig S2D-E: the observation that siRNA kd does not recapitulates the KO phenotype (gH2AX signals) is a significant concern (see my first point above).

Concerning the fact that siRNA-mediated NM1 knockdowns do not fully recapitulate the KO phenotype in the primary cells, it is important to remember that the latter are stable KO cells derived from an organism which has adapted to loss of NM1 whereas the former are transiently generated cells and may not have the time to fully respond to NM1 loss. In addition, siRNA knockdowns are generated by transfection of small interfering RNA oligonucleotides and because of the experimental nature of this approach the silencing method is never quantitative. In any case, as mentioned above we have now included NM1 KO cMEFs that recapitulate the phenotype observed in the primary embryonic fibroblasts.

Fig 4B: as some DNA repair genes are cell-cycle regulated, some sort of normalization on different proliferation rates/cell cycle is useful here.

We have now included FACS analysis of the cell cycle. The new results are part of figure 3 in the revised manuscript (Figure 3I).

Fig 3F: the observation that KO cells have more DNA damage and, despite that, proliferate more is perplexing and it is not “in line” or expected as the authors write

We have added a new proliferation experiment performed on NM1 KO cMEFs which is now part of figure 3 (Figure 3H) in the revised manuscript. This experiment confirms that in the absence of NM1 cells tend to proliferate at a faster pace. This observation together with the new FACS data (Figure 3I) supports the idea that loss of NM1 leads to a faster progression through the cell cycle. As mentioned in our first submission, this conclusion is compatible with the observation that p21 gene activity is downregulated. p21 is part of the p53-dependent checkpoint that is required to halt cell cycle progression in response to DNA damage. It is, therefore, expected that in the absence of p21 (downregulated in the NM1 KO cells) cell cycle progresses more rapidly skipping the DNA damage checkpoint.

Fig 5A needs IP in undamaged cells: otherwise the statement that such interaction is stimulated by DNA damage is not warranted

We have now included co-immunoprecipitations performed in the absence of DNA damage (see supplemental figure 2E). The results show that under these conditions NM1 can be co-precipitated with PCAF (as expected, see Sarshad et al 2013, 2014; Almuzzaini et al, 2015) but not with p53. This experiment, together with the co-IP performed under DNA damage conditions, supports the idea that NM1 and p53 interact upon DNA damage, although at this stage we cannot say that NM1 and p53 interact directly.

Minor:

Ref 18 is not a strong reference as heavy ions only were used.

Since this reference does not have any direct impact on our study or conclusions, we removed it from revised manuscript.

Reviewer #2 (Remarks to the Author)

This paper reports on the possible role of nuclear myosin I (NMI) in the DNA damage/ repair response. Since its discovery almost 20 years ago, progress on understanding the roles of NMI in the nucleus have been slower than on nuclear actin. In this respect, this paper is timely and of interest.

We thank this reviewer for the interest in our work.

My main issue with this paper is that it is descriptive and largely correlative in nature. Many pages in the Results section are devoted to listing possible interacting partners. While the RNA-Seq studies are important, at this stage they do not offer mechanistic insights into the role of NMI. There is an effort to discover mechanistic insights, but those data are not definitive. For instance, the observation that p21 and 14-3-3 are downregulated in NMI KO cells does not mean there is a direct line between NMI and the expression of these proteins. As pointed out on page 13, this pathway “may lead to impairment of cell cycle arrest” and most of the conclusions are qualified in a similar manner.

Although further work is required for in depth mechanistic analysis, we believe that our results do provide an initial glimpse on NM1 dependent activation of the transcriptional response to DNA damage, which is the main point of the paper. This conclusion directly comes from transcriptionally profiling primary cells lacking NM1 using RNAseq. Identification of the transcripts that are dysregulated indicates that genes upstream p53 dependent signaling are primarily upregulated whereas genes downstream p53 seem to be downregulated. In particular, we focused on p21 which is required to halt cell cycle progression in response to DNA damage. We show evidence that in WT cells NM1 binds to the transcription start site of the p21 gene and this leads to recruitment of PCAF and Set1 and opening up of the chromatin, a mechanism that is at the basis of p21 gene activation. In fact, in the absence of NM1, epigenetic marks for active transcription are lost and PCAF and Set1 are not recruited. Interestingly, binding of NM1 to the transcription start site of the p21 gene is independent of p53 and it happens in response to DNA damage. It is also interesting that upon DNA damage NM1 and p53 seem to interact (directly or indirectly is not known yet). On a speculative basis this suggests that there might be some structural changes to the chromatin that occurs upon DNA damage that ensure that p53 and NM1 become close enough to be able to interact. As part of a follow up study we are testing this model.

Concerning the suggestion that cell cycle may be affected, we have now included new data showing that, indeed, in the absence of NM1 we have a more rapid progression through the cell cycle precisely because p21 is down regulated (see figure 3H-I, in the revised manuscript).

On a separate note, we appreciate that 14-3-3 is also downregulated but in this study we focused on p21

There are also concerns regarding the knock out cells and the specificity of the antibody. Fig.

3A shows a blot in which the putative NMI band increases over time following etoposide treatment. Is this blot on KO cells or embryonic fibroblasts?

We believe that there is some misunderstanding here. As mentioned in the paper, this blot is on wild type embryonic fibroblasts - this is further clarified in the revised text.

How do these levels of NMI in KO cells compare to those in eto treated wild type cells?

Following up on the above, again there is some misunderstanding. The blots are on wild type embryonic fibroblasts subjected to etoposide treatment over a time course experiment. This blot indicates that over time there is an increase in NM1 expression in response to DNA damage. A comparison with the KO condition cannot be performed because NM1 is not expressed.

This is a little odd because there appears to be significant knock-down of NMI but the cells respond in a robust way to etoposide.

We do not agree with any of the results being at odd. Rather, this comment is unclear to us and probably results from the above misunderstanding by this reviewer.

Furthermore, if the antibodies cannot distinguish between NMI and cytoplasmic MI, how can one be sure this is truly due to an increase in NMI? One possibility is qPCR.

Again, concerning the specificity of the anti-NM1 antibody, there is a serious misunderstanding here. The anti-NM1 antibodies were generated in our lab years ago (see Fomproix and Percipalle, 2004). As can be seen in the original paper they are highly specific for NM1, raised against the unique N-terminal peptide exclusively present in NM1 and not in the other Myo1c isoforms. The antibody has been used in numerous applications, including immunoprecipitations and mass spectrometry (see for instance Sarshad et al, 2014 and supplementary results therein), immunofluorescence/confocal microscopy (for instance see Fomproix and Percipalle, 2004, Percipalle et al., 2006, EMBO Reports) as well as ChIPseq analysis (Almuzzaini et al., 2015). In any case to further emphasize the specificity of the anti-NM1 antibody, we have included a new immunoblot which is now part of figure 1 that shows an NM1 signal in wt MEFs but not in the KO condition. In conclusion, the specificity of the anti-NM1 antibody used in this study should not be a concern at all.

Regarding the knock outs, were the sequences used to target NMI specific to the 5' region unique to NMI or did they target the coding region that includes MI? The latter would knock down both and complicate the analysis because MI can substitute for NMI. If the 5' region of NMI was targeted, what was done to ensure the absence of off target effects? This region is rich in GC and such regions tend to be non-specific. Lastly, discrepancies in some of the data between cells derived from the knock out mice and th siRNA cells are disconcerting.

The NM1 KO mouse model was described years ago by the Hozak lab (see Venit et al, 2013), which is also included in the reference list of our manuscript). As can be seen in that study the KO is highly specific for NM1 and not for other forms of myosin 1. It was generated by the Cre/Lox system and not by CRISPR. Therefore, it is not possible that

there are off target effects as suggested by this reviewer. Since the primary embryonic fibroblasts used in this study are derived from the above mouse, we are confident that the KO is highly specific for NM1.

In regards to the disconcerting discrepancies detected by this reviewer between NM1 KO MEFs and siRNA-mediated NM1 knockdowns, it is important to remember that the former are stable KO cells i.e. derived from an organism which has adapted to loss of NM1 whereas the latter are transiently generated cells and may not have the time to fully respond to NM1 loss. In addition, siRNA knockdowns are generated by transfection of small interfering RNA oligonucleotides and because of the experimental nature of this approach the silencing method is never quantitative. In any case, to corroborate the observation that NM1 KO MEFs do display constitutive DNA damage foci, we have generated new NM1 KO MEFs by CRISPR. In the manuscript these cells are described as “NM1 KO cMEFs”. As can be seen in the revised version of the manuscript, analysis of these cells shows that they too display increased numbers of gamma H2AX positive foci compared to controls.

Other Points:

1. What statistical analyses were performed to test the significance of the data? Data in some of the figures are labelled statistically significant despite overlapping error bars. How was significance determined?

GraphPad Prism 8.3.0 software was used for statistical analysis. Unpaired t-test was used in all the experiments unless otherwise stated in the text. While error bars in bar charts represents standard deviation, the error bars in boxplots represent minimum and maximum values. Therefore, it could happen that in some graphs these error bars overlap. We now clearly stated this in the revised manuscript. We have also added a specific section on statistical analysis in the methods.

2. The Abstract states NMI KO cells exhibited increased proliferation. This appears to be based on Fig. 3F, which shows a barely significant different in growth rate at 24 hours only. This is not very convincing.

We have repeated the proliferation assay with NM1 WT and KO cMEFs (see figure 3H in the revised manuscript). Results from this experiment support changes in the proliferation rate in the absence of NM1. These results are corroborated by new FACS data (figure 3I) that support a faster progression through the cell cycle in the absence of NM1.

3. In Fig. 3A and S2D, is each spot a different cell nucleus? Please provide size markers.

Size markers have been added.

4. What cells were used in Fig. 3E? Are these total cell extracts or nuclear extracts?

As mentioned above, the cells are wild-type mouse embryonic fibroblasts. In these blots we used total cell extracts.

4. Fig. S2A: The DAPI panel does not seem to correlate with the other 2 panels. How many microns does the size marker represent?

We do not agree with this concern. DAPI staining does correlate with the other two panels. The discrepancy could happen because of reduced staining with NM1 antibody in some cells, which is expected as we were staining NM1 knock-down cells. The scale bar is 10 μ m. This figure is now labelled as supplementary figure 2F in the revised manuscript.

Reviewer #3 (Remarks to the Author):

Venit et al report a role for NM1 in supporting the p53-dependent induction of p21 following DNA damage. Phenotypic profiling and RNA-Seq of WT and KO MEFs suggest NM1 is important for generating histone marks associated with the induction of DNA damage/cell cycle regulator genes, notably p21. Consistent with these data, the authors find that NM1 is required for the recruitment of PCAF and SET1 to the p21 gene in response to etoposide.

This is potentially an interesting paper but lacks critical pieces of data. In addition, the results and discussion were extensively co-mingled. Even some figure legends were cast as discussion points. Significant editing and recasting will improve the presentation. Additional major comments are listed below:

1) Since NM1 KO mice are available, it is disappointing that the authors focused in MEFs as opposed to radiosensitive tissues. It may be that the analysis of tissues or a larger cohort of MEFs would strengthen the fairly weak PCA analysis in Figure S1 (WT3 is as far from WT1 and 2 as they are from the KO samples).

We agree with this reviewer that analysis of radiosensitive tissues would potentially provide interesting insights. However, given the complexity and the fact that multiple tissues would have to be compared, we believe that at this stage this goes beyond the scope of the present investigation but rather, a follow up study that we are currently pursuing.

2) The imaging data in Figure 1 should be confirmed by western blot for at least a subset of the histone marks. In addition an inordinate amount of text (nearly 3 pages!) was used to describe the initial imaging/RNA-Seq/bioinformatics analyses, most all of which was relegated to supplemental figures. The authors should consider placing Figure 1 in the supplemental figures and replacing it with a more informative figure composed of S1A, S1D and 2.

Thanks for pointing this out. We have now included a new immunoblot that corroborates the high content phenotypic profiling. As per suggestion by this reviewer, we have now shortened the text concerning initial imaging/RNAseq/bioinformatics. We appreciate the suggestion to reshuffle some of the figures, however we prefer to keep the quantitative chromatin data in figure 1 because it is the natural starting point of our story. We became interested in DNA damage genes because of the chromatin analysis and later on the fact that we discovered significant differential expression of genes involved in DNA damage and cell cycle.

3) The CTB proliferation data are not convincing and far less robust than the Myo1c data referred to by the authors. This shortcoming may result from the inefficient NM1 knockdown described in Figure S2A where only a subset of the cells appear to be transfected. See also point 6.

We have included new CTB proliferation data. This experiment was performed on new NM1 KO MEFs generated by CRISPR (described in the revised version as NM1 KO

cMEFs) and it is corroborated by FACS analysis of the cell cycle in the same cells which seem to progress through the cell cycle more rapidly.

4) Figure 4. Since the paper is centered on p21, it would be helpful to label the heatmap to identify which genes in these GO terms are regulated by or related to p21. It would also help the reader if the authors would indicate which part(s) of the 4A is analyzed in 4B and 4C, respectively.

All proteins from 4A are analyzed in 4B and 4C. We have clarified this in the revised manuscript. Proteins found in the intersection between cell cycle regulation and DNA damage response are plotted as a part of the figure 4B and are marked with a star. Proteins that are discussed in the paper in more detail and are interconnected with p21 are now marked with the black dot.

5) The co-IP data in Figure 5 do not show a complex; they only show a series of pairwise interactions. Additional separation methods would be required to suggest NM1, PCAF and p53 form a complex. The text on pages 14 and 15 needs to be revised to accordingly.

We agree that co-IP experiments do not indicate direct interactions or formation of a complex and we have rephrased the text.

6) Figure 5. The failure to induce p21 following etoposide treatment suggests that the NM1 deficient cells will fail to arrest. This possibility should be tested by flow analysis. Consequently, the NM1-deficient cells may well show increased apoptosis, which should also be examined. Correspondingly, the PCR analysis in panels B/D/F should be expanded to test whether NM1 loss affects induction of Puma or Bax

As mentioned earlier, we have now included new FACS data showing that the cell cycle progresses more rapidly in the absence of NM1. We have also tested apoptosis by FACS and as can be seen in the new panel shown in figure 3G, we detected increased apoptotic levels in NM1 KO cells.

Just to clarify, panels B/D/F in figure 5 of the manuscript show qPCR analysis of ChIP experiments using anti NM1, PCAF, Set1, p53, H3K9ac and H3K4me3 antibodies to show how occupancies change upon NM1 loss in response to DNA damage. In regards to testing the effects of NM1 loss on Puma or Bax levels, we checked it on the RNAseq data sets obtained in primary NM1 KO MEFs which are publicly available. We found that in the absence of NM1, Puma expression is not altered while Bax expression is significantly reduced. Although this observation is compatible with dysregulated p53 signaling, understanding the mechanisms would require considerably more work that goes beyond the scope of this study.

REVIEWERS' COMMENTS:

Reviewer #1 (Remarks to the Author):

The manuscript has now much improved and I am happy it is published as is.

Reviewer #2 (Remarks to the Author):

As I mentioned in my previous review, I think the subject is timely. It is, nevertheless, hard to follow because the authors seem to be undecided about their central message. The title includes p21 and the bottom line in Figure 6 is the regulation of p21 by NM1. But the Introduction mentions p21 almost as an afterthought and p21 seems to get buried in the body of the paper. The inability to decide on the central focus of this paper makes it very difficult to follow.

Furthermore, the main point of Figure 6 is the p21 expression is regulated via NM1 following DNA damage. In the absence of NM1, p21 expression is depressed and damaged DNA accumulates or is repaired less efficiently. It would have been very useful to demonstrate increases in p21 protein expression (ie: repeat the experiment in 5B with a western) and to demonstrate p21 expression is decreased in NM1 KO cells.

Potential off-target effects of knocking down NM1 remains an issue and the authors did not address my previous concerns. In light of the high G-C content of the 5' region of the NM1 gene, what experiments were performed to test for off target effects? It is very possible siRNA could have side effects and even CRISPR is not without drawbacks.

Regarding Fig. S2F (Formerly S2A), I am still confused. The bottom left panel (siRNA, DAPI) contains multiple, brightly DAPI stained nuclei that do not appear in the Merge. Also, what are the two very bright spots in the NM1, siRNA panel? Is this NM1 staining?

Etoposide induces apoptosis by blocking topoisomerase and creating double strand breaks. But Fig. 3E shows an increase in NM1 following etoposide treatment in wild type cells. In knock out cells, there is both an increase in the growth rate and the number of apoptotic cells. It is very difficult for me to visualize how this can happen simultaneously in the same cell. Incidentally, what concentration of etoposide used in this experiment?

The blot in Fig. 5A is of poor quality, especially the NM1 blot. The NM1 panels, except for the control IgG panel, contain two bands, a strong lower band and a weaker upper. In the absence of MW markers, which I requested in my previous review, it is difficult to know with certainty which is NM1 and which is cytoplasmic myosin 1.

Reviewer #3 (Remarks to the Author):

The revised manuscript is much improved. The authors were responsive to the original review, by including new and more compelling data and by significantly cleaning up the writing.

We would like to thank all reviewers for helping us improve our manuscript. Please find below our point-by-point responses (bold text) to each of the reviewers' concern.

Reviewer #1

The manuscript has now much improved and I am happy it is published as is.

We thank this reviewer for the positive evaluation of our manuscript.

Reviewer #2

Reviewer 2. As I mentioned in my previous review, I think the subject is timely. It is, nevertheless, hard to follow because the authors seem to be undecided about their central message. The title includes p21 and the bottom line in Figure 6 is the regulation of p21 by NM1. But the Introduction mentions p21 almost as an afterthought and p21 seems to get buried in the body of the paper. The inability to decide on the central focus of this paper makes it very difficult to follow.

The central message is the novel function of NM1 in the transcriptional response to DNA damage and therefore in genome stability. The introduction is divided into three main parts summarizing the known functions of Nuclear Myosin I (NM1), what is currently known about NM1 (and actin) function in DNA damage and the role of p21 in DNA damage as p53 target gene. In the last paragraph we summarize the findings of the study, highlighting the relation between NM1 and p21.

Based on the above, we believe that the introductory part on p21 is sufficiently focused and any additional information on p21 would be out of the scope of this paper, rather more suited for a general review of the field.

Reviewer 2. Furthermore, the main point of Figure 6 is the p21 expression is regulated via NM1 following DNA damage. In the absence of NM1, p21 expression is depressed and damaged DNA accumulates or is repaired less efficiently. It would have been very useful to demonstrate increases in p21 protein expression (ie: repeat the experiment in 5B with a western) and to demonstrate p21 expression is decreased in NM1 KO cells.

By directly measuring p21 RNA levels, we precisely show that NM1 affects transcription of the p21 gene. This conclusion is supported by both RT qPCR and RNAseq analyses which show that p21 RNA levels are significantly decreased in the NM1 KO conditions. These are quantitative experiments. Since we are monitoring transcript levels, we do not agree with this reviewer that Western blotting analyses would add any new information to the present study.

Reviewer 2. Potential off-target effects of knocking down NM1 remains an issue and the authors did not address my previous concerns. In light of the high G-C content of the 5' region of the NM1 gene, what experiments were performed to test for off target effects? It is very possible siRNA could have side effects and even CRISPR is not without drawbacks.

The primary cells used in the analysis have been derived from NM1 KO mice described before (see Venit at al., 2013). The knock-out mouse was prepared by conventional Cre/Lox system and not by CRISPR/Cas9 system. Cre/Lox system is based on homology recombination of long homology arms flanking around 2 LoxP sites surrounding gene of interest (or in this case 1st exon specific for NM1 expression) in embryonic cells and subsequent recombination by Cre recombinase which specifically recognize LoxP sites. Therefore, there is an extremely small possibility of having off-targets, something that was anyway tested in the original paper (Venit at al., 2013). To further strengthen our data, we prepared two new loss-of-function model systems for NM1 – one based on the CRISPR/Cas9 technology and the second one based on RNA interference. Even though we cannot exclude the possibility of off-targets in these systems, in both cases we observed the same phenotypes as those detected in primary knock-out cells. Therefore, we conclude that the observed phenotypes specifically originate from NM1 deletion/depletion.

Reviewer 2. Regarding Fig. S2F (Formerly S2A), I am still confused. The bottom left panel (siRNA, DAPI) contains multiple, brightly DAPI stained nuclei that do not appear in the Merge. Also, what are the two very bright spots in the NM1, siRNA panel? Is this NM1 staining?

As can be seen in the picture below, the first two frames are black/white pictures of DAPI and NM1 respectively which naturally seems brighter than the Merge panel using standard blue/green coloring on the black background. To facilitate this reviewer's interpretation of the data, we have included numbers to mark and co-localize each cell nucleus. As we are using siRNA to knockdown NM1, in the second panel we can see variability in NM1 expression (for example see cells 3,4,6,10 with very low NM1 expression and numbers 5,11,12 for higher NM1 expression). The two very bright spots mentioned by this reviewer (marked 14,15) are two newly made cells which presumably got out of mitosis and are likely to be in telophase/cytokinesis.

Reviewer 2. Etoposide induces apoptosis by blocking topoisomerase and creating double strand breaks. But Fig. 3E shows an increase in NM1 following etoposide treatment in wild type cells. In knock out cells, there is both an increase in the growth rate and the number of apoptotic cells. It is very difficult for me to visualize how this can happen simultaneously in the same cell. Incidentally, what concentration of etoposide used in this experiment?

We have shown that NM1 expression is increased in WT cells upon induction of DNA damage and that NM1 KO cells have higher proliferation and apoptosis. These two experiments are not in contrast with

each other as suggested by this reviewer. Rather, these experiments complement each other. In WT cells, since NM1 is part of the cellular response to DNA damage, NM1 expression levels increase concomitantly with the extent of etoposide induced DNA damage. This leads to p21 gene activation and consequently, cell cycle arrest. However, upon NM1 KO, cells have lost their ability to respond to DNA damage, proceed through the cell cycle in an uncontrolled manner (indeed, we detect increased proliferation rates) and accumulate DNA damage through several generations to reach the point when they continue towards programmed cell death (more apoptosis). Etoposide concentration is described in Material and methods part for each experiment.

Reviewer 2. The blot in Fig. 5A is of poor quality, especially the NM1 blot. The NM1 panels, except for the control IgG panel, contain two bands, a strong lower band and a weaker upper. In the absence of MW markers, which I requested in my previous review, it is difficult to know with certainty which is NM1 and which is cytoplasmic myosin 1.

While we partly agree with this reviewer about the quality of the blots in fig 5A, it is important to compare the NM1-IP with the IgG-IP (control). As can be seen, NM1 is not co-precipitated with the IgG and this strongly support the specificity of the antibody and thus, the specificity of the entire experiment. We would also like to point out that the specificity of the anti-NM1 antibody should not be a concern. This anti-NM1 antibody has been previously used in several studies by our lab as well as other labs and it specifically recognizes the N-terminal 16 amino acid uniquely found in NM1 and not in the other Myo1C isoforms.

Molecular weight markers are now included in the revised figure Fig5A. However, we would like to point out that the difference in molecular weight between Myo1C and NM1 is difficult to appreciate by gel electrophoresis and western blotting since their molecular weights differ by 2 KDa (118 kDa for Myo1C and 120kDa NM1).

Reviewer #3

The revised manuscript is much improved. The authors were responsive to the original review, by including new and more compelling data and by significantly cleaning up the writing.

We thank this reviewer for the constructive comments and for helping us improve the manuscript.